# LAVA: Data Valuation without Pre-Specified Learning Algorithms

Hoang Anh Just [*1], Feiyang Kang [*1], Jiachen T. Wang[2], Yi Zeng[1], Myeongseob Ko[1],
Ming Jin[1], and Ruoxi Jia[1]

[1]Virginia Tech, [2]Princeton University
{just, fyk, yizeng, myeongseob, jinming, ruoxijia}@vt.edu
tianhaowang@princeton.edu

## Abstract

Traditionally, data valuation is posed as a problem of equitably splitting the validation performance of a learning algorithm among the training data. As a result, the calculated data values depend on many design choices of the underlying learning algorithm. However, this dependence is undesirable for many use cases of data valuation, such as setting priorities over different data sources in a data acquisition process and informing pricing mechanisms in a data marketplace. In these scenarios, data needs to be valued before the actual analysis and the choice of the learning algorithm is still undetermined then. Another side-effect of the dependence is that to assess the value of individual points, one needs to re-run the learning algorithm with and without a point, which incurs a large computation burden.

This work leapfrogs over the current limits of data valuation methods by introducing a new framework that can value training data in a way that is oblivious to the downstream learning algorithm. Our main results are as follows. **(1)** We develop a proxy for the validation performance associated with a training set based on a non-conventional *class-wise Wasserstein distance* between the training and the validation set. We show that the distance characterizes the upper bound of the validation performance for any given model under certain Lipschitz conditions. **(2)** We develop a novel method to value individual data based on the sensitivity analysis of the class-wise Wasserstein distance. Importantly, these values can be directly obtained *for free* from the output of off-the-shelf optimization solvers when computing the distance. **(3)** We evaluate our new data valuation framework over various use cases related to detecting low-quality data and show that, surprisingly, the learning-agnostic feature of our framework enables a significant improvement over the state-of-the-art performance while being orders of magnitude faster.

## 1 Introduction

Advances in machine learning (ML) crucially rely on the availability of large, relevant, and high-quality datasets. However, real-world data sources often come in different sizes, relevance levels, and qualities, differing in their value for an ML task. Hence, a fundamental question is how to quantify the value of individual data sources. Data valuation has a wide range of use cases both within the domain of ML and beyond. It can help practitioners enhance the model performance through prioritizing high-value data sources (Ghorbani & Zou, 2019), and it allows one to make strategic and economic decisions in data exchange (Scelta et al., 2019).

In the past literature (Ghorbani & Zou, 2019; Jia et al., 2019b; Kwon & Zou, 2021), data valuation is posed as a problem of equitably splitting the validation performance of a given learning algorithm among the training data. Formally, given a training dataset $\mathbb{D}_t = \{z_i\}_{i=1}^{N}$, a validation dataset $\mathbb{D}_v$, a learning algorithm $\mathcal{A}$, and a model performance metric PERF (e.g., classification accuracy), a *utility function* is first defined over all subsets $S \subseteq \mathbb{D}_t$ of the training data: $U(S) := \text{PERF}(\mathcal{A}(S))$. Then, the objective of data valuation is to find a score vector $s \in \mathbb{R}^N$ that represents the allocation to each datapoint. For instance, one simple way to value a point $z_i$ is through leave-one-out (LOO) error $U(\mathbb{D}_t) - U(\mathbb{D}_t \setminus \{z_i\})$, i.e., the change of model performance when the point is excluded from training. Most of the recent works have leveraged concepts originating from cooperative game theory (CGT), such as the Shapley value (Ghorbani & Zou, 2019; Jia et al., 2019b), Banzhaf value (Wang

---

[*]Equal contribution. Repository publicly available on Github: https://github.com/ruoxi-jia-group/LAVA.

& Jia, 2022), general semivalues (Kwon & Zou, 2021), and Least cores (Yan & Procaccia, 2021) to value data. Like the LOO, all of these concepts are defined based on the utility function.

Since the utility function is defined w.r.t. a specific learning algorithm, the data values calculated from the utility function also depend on the learning algorithm. In practice, there are many choice points pertaining to a learning algorithm, such as the model to be trained, the type of learning algorithm, as well as the hyperparameters. The detailed settings of the learning algorithms are often derived from data analysis. However, in many critical applications of data valuation such as informing data acquisition priorities and designing data pricing mechanism, data needs to be valued before the actual analysis and the choice points of the learning algorithm are still undetermined at that time. This gap presents a main hurdle for deploying existing data valuation schemes in the real world.

The reliance on learning algorithms also makes existing data valuation schemes difficult to scale to large datasets. The exact evaluation of LOO error and CGT-based data value notions require evaluating utility functions over different subsets and each evaluation entails retraining the model on that subset: the number of retraining times is linear in the number of data points for the former, and exponential for the latter. While existing works have proposed a variety of approximation algorithms, scaling up the calculation of these notions to large datasets remains expensive. Further, learning-algorithm-dependent approaches rely on the performance scores associated with models trained on different subsets to determine the value of data; thus, they are susceptible to noise due to training stochasticity when the learning algorithm is randomized (e.g., SGD) (Wang & Jia, 2022).

This work addresses these limitations by introducing a *learning-agnostic* data **va**luation (LAVA) framework. LAVA is able to produce efficient and useful estimates of data value in a way that is oblivious to downstream learning algorithms. Our technical contributions are listed as follows.

**Proxy for validation performance.** We propose a proxy for the validation performance associated with a training set based on the non-conventional class-wise Wasserstein distance (Alvarez-Melis & Fusi, 2020) between the training and the validation set. The hierarchically-defined Wasserstein distance utilizes a hybrid Euclidean-Wasserstein cost function to compare the feature-label pairs across datasets. We show that this distance characterizes the upper bound of the validation performance of any given models under certain Lipschitz conditions.

**Sensitivity-analysis-based data valuation.** We develop a method to assess the value of an individual training point by analyzing the sensitivity of the particular Wasserstein distance to the perturbations on the corresponding probability mass. The values can be directly obtained *for free* from the output of off-the-shelf optimization solvers once the Wasserstein distance is computed. As the Wasserstein distance can be solved much more efficiently with entropy regularization (Cuturi, 2013), in our experiments, we utilize the duals of the entropy-regularized program to approximate the sensitivity. Remarkably, we show that the gap between two data values under the original non-regularized Wasserstein distance can be recovered *exactly* from the solutions to the regularized program.

**State-of-the-art performance for differentiating data quality.** We evaluate LAVA over a wide range of use cases, including detecting mislabeled data, backdoor attacks, poisoning attacks, noisy features, and task-irrelevant data, in which some of these are first conducted in the data valuation setting. Our results show that, surprisingly, the learning-agnostic feature of our framework enables a significant performance improvement over existing methods, while being orders of magnitude faster.

## 2 MEASURING DATASET UTILITY VIA OPTIMAL TRANSPORT

In this section, we consider the problem of quantifying training data utility $U(\mathbb{D}_t)$ without the knowledge of learning algorithms. Similar to most of the existing data valuation frameworks, we assume access to a set of validation points $\mathbb{D}_v$. Our idea is inspired by recent work on using the *hierarchically-defined Wasserstein distance* to characterize the relatedness of two datasets (Alvarez-Melis & Fusi, 2020). Our contribution here is to apply that particular Wasserstein distance to the data valuation problem and provide a theoretical result that connects the distance to validation performance of a model, which might be of independent interest.

### 2.1 OPTIMAL TRANSPORT-BASED DATASET DISTANCE

**Background on Optimal Transport (OT).** OT is a celebrated choice for measuring the discrepancy between probability distributions (Villani, 2009). Compared to other notable dissimilarity measures such as the Kullback-Leibler Divergence (Kullback & Leibler, 1951) or Maximum Mean Discrepan-

cies (MMD) (Szekely et al., 2005), the mathematically well-defined OT distance has advantageous analytical properties. For instance, OT is a distance metric, being computationally tractable and computable from finite samples (Genevay et al., 2018; Feydy et al., 2019).

The Kantorovich formulation (Kantorovich, 1942) defines the OT problem as a Linear Program (LP). Given probability measures $\mu_t, \mu_v$ over the space $\mathcal{Z}$, the OT problem is defined as $\mathrm{OT}(\mu_t, \mu_v) := \min_{\pi \in \Pi(\mu_t, \mu_v)} \int_{\mathcal{Z}^2} \mathcal{C}(z, z') d\pi(z, z')$ where $\Pi(\mu_t, \mu_v) := \left\{ \pi \in \mathcal{P}(\mathcal{Z} \times \mathcal{Z}) \mid \int_{\mathcal{Z}} \pi(z, z') dz = \mu_t, \int_{\mathcal{Z}} \pi(z, z') dz' = \mu_v \right\}$ denotes a collection of couplings between two distributions $\mu_t$ and $\mu_v$ and $\mathcal{C} : \mathcal{Z} \times \mathcal{Z} \to \mathbb{R}^+$ is some symmetric positive cost function (with $\mathcal{C}(z, z) = 0$), respectively. If $\mathcal{C}(z, z')$ is the Euclidean distance between $z$ and $z'$ according to the distance metric d, then $\mathrm{OT}(\mu_t, \mu_v)$ is 2-Wasserstein distance, which we denote as $W_{\mathcal{C}}(\mu_t, \mu_v) = W_{\mathrm{d}}(\mu_t, \mu_v) := \mathrm{OT}(\mu_t, \mu_v)$. In this work, the notation OT and $W$ are used interchangeably, with a slight difference that we use OT to emphasize various of its formulations while $W$ specifies on which distance metric it is computed.

**Measuring Dataset Distance.** We consider a multi-label setting where we denote $f_t : \mathcal{X} \to \{0, 1\}^V$, $f_v : \mathcal{X} \to \{0, 1\}^V$ as the labeling functions for training and validation data, respectively, where $V$ is the number of different labels. Given the training set $\mathbb{D}_t = \{(x_i, f_t(x_i))\}_{i=1}^N$ of size $N$, and the validation set $\mathbb{D}_v = \{(x_i', f_v(x_i'))\}_{i=1}^M$ of size $M$, one can construct discrete measures $\mu_t(x, y) := \frac{1}{N} \sum_{i=1}^N \delta_{(x_i, y_i)}$ and $\mu_v(x, y) := \frac{1}{M} \sum_{i=1}^M \delta_{(x_i', y_i')}$, where $\delta$ is Dirac function. Consider that each datapoint consists of a feature-label pair $(x_i, y_i) \in \mathcal{X} \times \mathcal{Y}$. While the Euclidean distance naturally provides the metric to measure distance between features, the distance between labels generally lacks a definition. Consequently, we define conditional distributions $\mu_t(x|y) := \frac{\mu_t(x) I[f_t(x) = y]}{\int \mu_t(x) I[f_t(x) = y] dx}$ and $\mu_v(x|y) := \frac{\mu_v(x) I[f_v(x) = y]}{\int \mu_v(x) I[f_v(x) = y] dx}$. Inspired by Alvarez-Melis & Fusi (2020), we measure the distance between two labels in terms of the OT distance between the conditional distributions of the features given each label. Formally, we adopt the following cost function between feature-label pairs: $\mathcal{C}((x_t, y_t), (x_v, y_v)) := \mathrm{d}(x_t, x_v) + c W_{\mathrm{d}}(\mu_t(\cdot|y_t), \mu_v(\cdot|y_v))$, where $c \geq 0$ is a weight coefficient. We note that $\mathcal{C}$ is a distance metric since $W_{\mathrm{d}}$ is a valid distance metric. With the definition of $\mathcal{C}$, we propose to measure the distance between the training and validation sets using the non-conventional, hierarchically-defined Wasserstein distance between the corresponding discrete measures: $W_{\mathcal{C}}(\mu_t, \mu_v) = \min_{\pi \in \Pi(\mu_t, \mu_v)} \int_{\mathcal{Z}^2} \mathcal{C}(z, z') d\pi(z, z')$.

Despite its usefulness and potentially broad applications, we note that it remains absent for existing research to explore its theoretical properties or establish applications upon this notion. This work aims to fill this gap by extending in both directions–novel analytical results are presented to provide its theoretical justifications while an original computing framework is proposed that extends its applications to a new scenario of datapoint valuation.

**Computational Acceleration via Entropic Regularization.** Solving the problem above scales cubically with $MN$, which is prohibitive for large datasets. Entropy-regularized OT (entropy-OT) becomes a prevailing choice for approximating OT distances as it allows for fastest-known algorithms. Using the iterative Sinkhorn algorithm (Cuturi, 2013) with almost linear time complexity and memory overhead, entropy-OT can be implemented on a large scale with parallel computing (Genevay et al., 2018; Feydy et al., 2019). Given a regularization parameter, $\varepsilon > 0$, entropy-OT can be formulated as $\mathrm{OT}_\varepsilon(\mu_t, \mu_v) := \min_{\pi \in \Pi(\mu_t, \mu_v)} \int_{\mathcal{Z}^2} \mathcal{C}(z, z') d\pi(z, z') + \varepsilon H(\pi | \mu_t \otimes \mu_v)$, where $H(\pi | \mu_t \otimes \mu_v) = \int_{\mathcal{Z}^2} \log \left( \frac{d\pi}{d\mu_t d\mu_v} \right) d\pi$. As $\varepsilon \to 0$, the dual solutions to the $\varepsilon$-entropy-OT converge to its OT counterparts as long as the latter are unique (Nutz & Wiesel, 2021).

## 2.2 LOWER *Class-Wise Wasserstein Distance* ENTAILS BETTER VALIDATION PERFORMANCE

In this paper, we propose to use $W_{\mathcal{C}}$, a *non-conventional, class-wise Wasserstein distance* w.r.t. the special distance function $\mathcal{C}$ defined in 2.1, as a learning-agnostic surrogate of validation performance to measure the utility of training data. Note that while Wasserstein distances have been frequently used to bound the learning performance change due to distribution drift (Courty et al., 2017; Damodaran et al., 2018; Shen et al., 2018; Ge et al., 2021), this paper is the first to bound the performance change by the hierarchically-defined Wasserstein distance with respect to the hybrid cost $\mathcal{C}$.

Figure 1 provides an empirical justification for using this novel distance metric as a proxy, and presents a relation between the class-wise Wasserstein distance and a model's validation performance. Each curve represents a certain dataset trained on a specific model to receive its performance. Since,

each dataset is of different size and structure, their distances will be of different scale. Therefore, we normalize the distances to the same scale to present the relation between the Wasserstein distance and model performance, which shows that despite different datasets and models, with increased distance, the validation performance decreases.

The next theorem theoretically justifies using this Wasserstein distance as a proxy for validation performance of a model. With assumptions on Lipschitzness of the downstream model as well as the labeling functions associated with the training and validation sets (as explicated in Appendix A), we show that the discrepancy between the training and validation performance of a model is bounded by the hierarchically-defined Wasserstein distance between the training and the validation datasets.

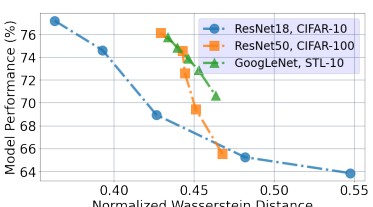

Figure 1: Normalized Wasserstein distance vs. model performance on different datasets and models.

**Theorem 1.** *We denote $f_t : \mathcal{X} \to \{0,1\}^V$, $f_v : \mathcal{X} \to \{0,1\}^V$ as the labeling functions for training and validation data, where $V$ is the number of different labels. Let $f : \mathcal{X} \to [0,1]^V$ be the model trained on training data. By definitions, we have that $\|f(\cdot)\|, \|f_t(\cdot)\|, \|f_v(\cdot)\| \leq V$. Let $\mu_t$, $\mu_v$ be the training and validation distributions, respectively, and let $\mu_t(\cdot|y)$ and $\mu_v(\cdot|y)$ be the corresponding conditional distributions given label $y$. Assume that the model $f$ is $\epsilon$-Lipschitz and the loss function $\mathcal{L} : \{0,1\}^V \times [0,1]^V \to \mathbb{R}^+$ is $k$-Lipschitz in both inputs. Define cost function $\mathcal{C}$ between $(x_v, y_v)$ and $(x_t, y_t)$ as $\mathcal{C}((x_t, y_t), (x_v, y_v)) := \mathrm{d}(x_t, x_v) + cW_{\mathrm{d}}(\mu_t(\cdot|y_t), \mu_v(\cdot|y_v))$, where $c$ is a constant. Under a certain cross-Lipschitzness assumption for $f_t$ and $f_v$ detailed in Appendix A, we have $\mathbb{E}_{x \sim \mu_v(x)} [\mathcal{L}(f_v(x), f(x))] \leq \mathbb{E}_{x \sim \mu_t(x)} [\mathcal{L}(f_t(x), f(x))] + k\epsilon W_{\mathcal{C}}(\mu_t, \mu_v) + \mathcal{O}(kV)$.*

Proofs are deferred to Appendix A. The bound is interesting to interpret. The first term on the right-hand side corresponds to the training performance. In practice, when a model with large enough capacity is used, this term is small. The second one is the exact expression of the Wasserstein distance that we propose to use as a proxy for validation performance. The last error term is due to possible violation of the cross-Lipschitzness assumption for $f_t$ and $f_v$. This term will be small if $f_t$ and $f_v$ assign the same label to close features with high probability. If the last term is small enough, it is possible to use the proposed Wasserstein distance as proxy for validation loss provided that $f$, $f_t$ and $f_v$ verify the cross-Lipschitz assumptions. The bound resonates with the empirical observation in Figure 1 that with lower distance between the training and the validation data, the validation loss of the trained model decreases.

## 3 EFFICIENT VALUATION OF INDIVIDUAL DATAPOINTS

Note that the class-wise Wasserstein distance defined in the previous section can be used to measure the utility for subsets of $\mathbb{D}_t$. Given this utility function, one can potentially use existing CGT-based notions such as the Shapley value to measure the contribution of individual points. However, even approximating these notions requires evaluating the utility function on a large number of subsets, which incurs large extra computation costs. In this section, we introduce a new approach to valuating individual points. Remarkably, our values can be directly obtained *for free* from the output of off-the-shelf optimization solvers once the proposed Wasserstein distance between the full training and testing datasets is computed.

### 3.1 DATAPOINT VALUATION VIA PARAMETER SENSITIVITY

OT distance is known to be insensitive to small differences while also being not robust to large deviations (Villani, 2021). This feature is naturally suitable for detecting abnormal datapoints—disregarding normal variations in distances between clean data while being sensitive to abnormal distances of outlying points. We propose to measure individual points' contribution based on the gradient of the OT distance to perturbations on the probability mass associated with each point.

Gradients are local information. However, unlike widely used influence functions that only hold for infinitesimal perturbation (Koh & Liang, 2017), gradients for LP hold precisely in a local range and still encode partial information beyond that range, making it capable of reliably predicting the change to the OT distance due to adding or removing datapoints without the need of re-calculation. Also, the gradients are directed information, revealing both positive and negative contributions for each

datapoint and allowing one to perform ranking of datapoints based on the gradient values. Finally, the OT distance always considers the collective effect of all datapoints in the dataset.

Leveraging the duality theorem for LP, we rewrite the *original* OT problem (introduced in 2.1) in the equivalent form: $\mathrm{OT}(\mu_t, \mu_v) := \max_{(f,g) \in C^0(\mathcal{Z})^2} \langle f, \mu_t \rangle + \langle g, \mu_v \rangle$, where $C^0(\mathcal{Z})$ is the set of all continuous functions, $f$ and $g$ are the dual variables. Let $\pi^*$ and $(f^*, g^*)$ be the corresponding optimal solutions to the primal and dual problems. The Strong Duality Theorem indicates that $\mathrm{OT}(\pi^*(\mu_t, \mu_v)) = \mathrm{OT}(f^*, g^*)$, where the right-hand side is the distance parameterized by $\mu_t$ and $\mu_v$. From the Sensitivity Theorem (Bertsekas, 1997), we have that the gradient of the distance w.r.t. the probability mass of datapoints in the two datasets can be expressed as follows: $\nabla_{\mu_t} \mathrm{OT}(f^*, g^*) = (f^*)^T$, $\nabla_{\mu_v} \mathrm{OT}(f^*, g^*) = (g^*)^T$. Note that the original formulation in 2.1 is always redundant as the constraint $\sum_{i=1}^{N} \mu_t(z_i) = \sum_{i=1}^{M} \mu_v(z_i') = 1$ is already implied, rendering the dual solution to be non-unique. To address this issue, we first remove any one of the constraints in $\Pi(\mu_t, \mu_v)$ and make the primal formulation non-degenerate. Then, we assign a value of zero to the dual variable corresponding to that removed primal constraint.

When measuring the gradients of the OT distance w.r.t. the probability mass of a given datapoint in each dataset, we calculate the *calibrated gradient* as

$$\frac{\partial \mathrm{OT}(\mu_t, \mu_v)}{\partial \mu_t(z_i)} = f_i^* - \sum_{j \in \{1,\dots N\} \backslash i} \frac{f_j^*}{N-1}, \quad \frac{\partial \mathrm{OT}(\mu_t, \mu_v)}{\partial \mu_v(z_i')} = g_i^* - \sum_{j \in \{1,\dots M\} \backslash i} \frac{g_j^*}{M-1}, \quad (1)$$

which represents the rate of change in the OT distance w.r.t the change of the probability mass of a given datapoint *along the direction* ensuring the probability mass for all datapoints in the dataset always sums up to one (explicitly enforcing the removed constraint). The value of calibrated gradients is independent of the choice of selection during the constraint removal.

**Datapoint valuation via calibrated gradients.** The calibrated gradients predict how the OT distance changes as more probability mass is shifted to a given datapoint. This can be interpreted as a measure of the contribution of the datapoint to the OT distance. The contribution can be positive or negative, suggesting shifting more probability mass to this datapoint would result in an increase or decrease of the dataset distance, respectively. If we want a training set to match the distribution of the validation dataset, then removing datapoints with large positive gradients while increasing datapoints with large negative gradients can be expected to reduce their OT distance. As we will show later, the calibrated gradients can provide a tool to detect abnormal or irrelevant data in various applications.

**Radius for accurate predictions.** The Linear Programming theories (Bertsimas & Tsitsiklis, 1997) give that for each non-degenerate optimal solution, we are always able to perturb parameters on the right-hand side of primal constraints ($\Pi(\mu_t, \mu_v)$ in 2.1) in a small range without affecting the optimal solution to the dual problem. When the perturbation goes beyond a certain range, the dual solution becomes primal infeasible and the optimization problem needs to be solved again. Hence, the calibrated gradients are local information and we would like to know the perturbation radius such that the optimal dual solution remains unchanged—i.e., whether this range is large enough such that the calibrated gradients can accurately predict the *actual* change to the OT distance. If the perturbation goes beyond this range, the prediction may become inaccurate as the dual solution only encodes partial information about the optimization.

In our evaluation, we find that this range is about 5% to 25% of the probability measure of the datapoint ($\mu_{(\cdot)}(z_i)$) for perturbations in both directions and the pattern seems independent of the size of the datasets. This range being less than the probability mass of a datapoint suggests that we are only able to predict the change to the OT distance for removing/adding a datapoint to the dataset approximately, though, the relative error is well acceptable (depicted in Figure 2).

### 3.2 PRECISE RECOVERY OF RANKING FOR DATA VALUES OBTAINED FROM ENTROPY-OT

Due to computational advantages of the entropy-OT (defined in Eq. 2.1), one needs to resort to the solutions to entropy-OT to calculate data values. We quantify the deviation in the calibrated gradients caused by the entropy regularizer. This analysis provides foundations on the potential impact of the deviation on the applications built on these gradients.

**Theorem 2.** *Let* $\mathrm{OT}(\mu_t, \mu_v)$ *and* $\mathrm{OT}_\varepsilon(\mu_t, \mu_v)$ *be the original formulation and entropy penalized formulation (as defined in 2.1) for the OT problem between the empirical measures* $\mu_t$ *and* $\mu_v$ *associated with the two datasets* $\mathbb{D}_t$ *and* $\mathbb{D}_v$, *respectively, where* $|\mathbb{D}_t| = N$ *and* $|\mathbb{D}_v| = M$. *Then,*

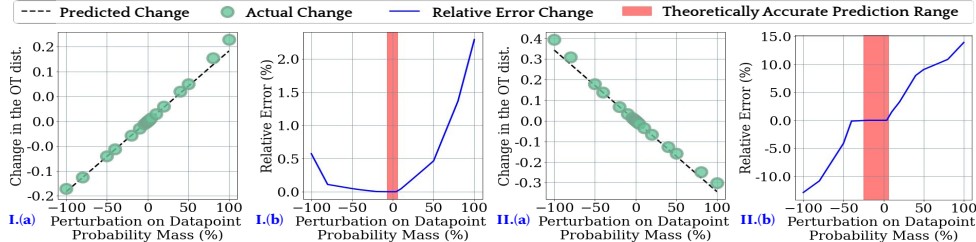

Figure 2: Predicting the change to the OT distance for increasing/reducing the probability mass of a point. The OT distance is calculated between two subsets of CIFAR-10. We examine the change to the OT distance predicted by the calibrated gradients against the actual change. The results of two datapoints are visualized for demonstration (I.(a)/(b) and II.(a)/(b) are analyzed on datapoint #1 and #2, respectively). The probability mass of the datapoint is perturbed from $-100\%$ (removing the datapoint) to $100\%$ (duplicating the datapoint). I.(a) and II.(a): Predicted change on the OT distance against the actual change. The predicted change demonstrated high consistency to the actual change despite minor deviation for large perturbation. I.(b) and II.(b): Relative error for the prediction, defined as (predicted_change - actual_change)/actual_change$\times 100\%$. The color bar represents the theoretical range of perturbation where the change in the OT distance can be accurately predicted. The prediction holds approximately well beyond the range.

*for any $i \neq j \neq k \in \{1, 2, \ldots, N\}$ and $o \neq p \neq q \in \{1, 2, \ldots, M\}$, the difference between the calibrated gradients for two datapoints $z_i$ and $z_k$ in dataset $\mathbb{D}_t$ and the difference for $z'_p$ and $z'_q$ in $\mathbb{D}_v$ can be calculated as*

$$\frac{\partial \text{OT}(\mu_t, \mu_v)}{\partial \mu_t(z_i)} - \frac{\partial \text{OT}(\mu_t, \mu_v)}{\partial \mu_t(z_k)} = \frac{\partial \text{OT}_\varepsilon(\mu_t, \mu_v)}{\partial \mu_t(z_i)} - \frac{\partial \text{OT}_\varepsilon(\mu_t, \mu_v)}{\partial \mu_t(z_k)} - \varepsilon \cdot \frac{N}{N-1} \cdot \left( \frac{1}{(\pi_\varepsilon^*)_{kj}} - \frac{1}{(\pi_\varepsilon^*)_{ij}} \right), \quad (2)$$

$$\frac{\partial \text{OT}(\mu_t, \mu_v)}{\partial \mu_v(z'_p)} - \frac{\partial \text{OT}(\mu_t, \mu_v)}{\partial \mu_v(z'_q)} = \frac{\partial \text{OT}_\varepsilon(\mu_t, \mu_v)}{\partial \mu_v(z'_p)} - \frac{\partial \text{OT}_\varepsilon(\mu_t, \mu_v)}{\partial \mu_v(z'_q)} - \varepsilon \cdot \frac{M}{M-1} \cdot \left( \frac{1}{(\pi_\varepsilon^*)_{qo}} - \frac{1}{(\pi_\varepsilon^*)_{po}} \right), \quad (3)$$

*where $\pi_\varepsilon^*$ is the optimal primal solution to the entropy penalized OT problem defined in 2.1, $z_j$ is any datapoint in $\mathbb{D}_t$ other than $z_i$ or $z_k$, and $z'_o$ is any datapoint in $\mathbb{D}_v$ other than $z'_p$ or $z'_q$.*

The gradient difference on the left-hand side of (2) represents the *groundtruth* value difference between two training points $z_i$ and $z_k$ as the values are calculated based on the original OT formulation. In practice, for the sake of efficiency, one only solves the regularized formulation instead and, therefore, this groundtruth difference cannot be obtained directly. Theorem 2 nevertheless indicates a very interesting fact that one can calculate the groundtruth difference based on the solutions to the regularized problem, because every term in the right-hand side only depends on the solutions to the regularized problem. Particularly, the groundtruth value difference is equal to the value difference produced by the regularized solutions plus some calibration terms that scale with $\varepsilon$ (Nutz & Wiesel, 2021). This result indicates that while it is not possible to obtain *individual groundtruth value* by solving the regularized problem, one can actually *exactly* recover the groundtruth value *difference* based on the regularized solutions. In many applications of data valuation such as data selection, it is the order of data values that matters (Kwon & Zou, 2021). For instance, to filter out low-quality data, one would first rank the datapoints based on their values and then throw the points with lowest values. In these applications, solving the entropy-regularized program is an ideal choice—which is both efficient and recovers the exact ranking of datapoint values. Finally, note that Eq. 3 presents a symmetric result for the calibrated gradients for validation data. In our experiments, we set $\epsilon = 0.1$, rendering the corresponding calibration terms to be negligible. As a result, we can directly use the calibrated gradients solved by the regularized program to rank datapoint values.

## 4 EXPERIMENTS

In this section, we demonstrate the practical efficacy and efficiency of LAVA on various classification datasets. We compare with nine baselines: (1) Influence functions (INF) (Koh & Liang, 2017), which approximates the LOO error with first-order extrapolation; (2) TracIn-Clean (Pruthi et al., 2020), which accumulates the loss change on validation data during training whenever the training point of interest is sampled; (3) TracIn-Self (Pruthi et al., 2020), which is similar to TracIn-Clean but accumulates the training loss changes; (4) KNN-Shapley (KNN-SV) (Jia et al., 2019a), which

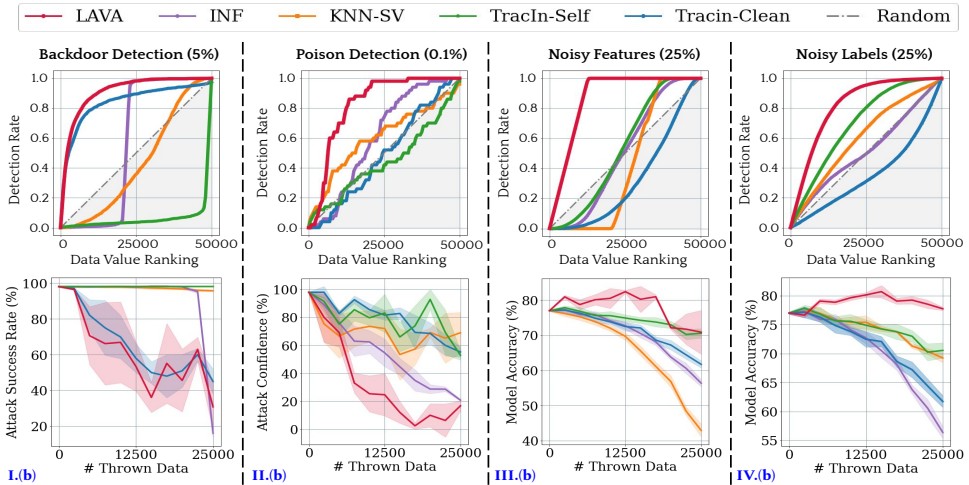

Figure 3: Performance comparison between LAVA and baselines on various use cases. For I.(b), we depict the Attack Accuracy, where lower value indicates more effective detection. For II.(b), we depict the Attack Confidence, as lower confidence indicates better poison removal. For III.(b) and IV.(b), we show the model test accuracy, where higher accuracy means effective data removal.

approximates the Shapley value using K-Nearest-Neighbor as a proxy model; and (5) Random, a setting where we select a random subset from the target dataset. We also consider the popular data valuation approaches: (6) Permutation Sampling-based Shapely value (Perm-SV) (Jia et al., 2019b), (7) Least Cores (LC) (Yan & Procaccia, 2021), (8) TMC-Shapley (TMC-SV) and (9) G-Shapley (G-SV) (Ghorbani & Zou, 2019). Baselines (6)-(9) are, however, computationally infeasible for the scale of data that we study here. So we exclude them from the evaluation of efficacy in different use cases. We also provide a detailed runtime comparison of all baselines. For all methods to be compared, a validation set of $10,000$ samples is assumed. For our method, we first use the validation data to train a deep neural network model PreActResNet18 (He et al., 2016) from scratch for feature extraction. Then, from its output, we compute the class-wise Wasserstein distance and the calibrated gradients for data valuation. Details about datasets, models, hyperparameter settings, and ablation studies of the hyperparameters and validation sizes are provided in Appendix B.

We evaluate on five different use cases of data valuation: *detecting backdoor attack, poisoning attack, noisy features, mislabeled data, and irrelevant data*. The first four are conventional tasks in the literature and the last one is a new case. All of them have a common goal of identifying "low-quality" training points. To achieve this goal, we rank datapoints in ascending order of their values and remove some number of points with lowest data values. For each removal budget, we calculate the *detection rate*, i.e., the percentage of the points that are truly bad within the removed points.

**Backdoor Attack Detection.** A popular technique of introducing backdoors to models is by injecting maliciously constructed data into a training set (Zeng et al., 2021). At test time, any trained model would misclassify inputs patched with a backdoor trigger as the adversarially-desired target class. In the main text, we consider the Trojan Square attack, a popular attack algorithm (Liu et al., 2017), which injects training points that contain a backdoor trigger and are relabeled as a target class. The evaluation of other types of backdoor attacks can be found in Appendix B. To simulate this attack, we select the target attack class *Airplane* and poison 2500 $(5\%)$ samples of the total CIFAR-10 training set $(50k)$ with a square trigger. In Figure 3 I.(a), we compare the detection rates of different data valuation methods. LAVA and TracIn-Clean outperform the others by a large margin. In particular, for LAVA, the first $20\%$ of the points that it removes contain at least $80\%$ of the poisoned data. We also evaluate whether the model trained after the removal still suffers from the backdoor vulnerability. To perform this evaluation, we calculate the *attack accuracy*, i.e., the accuracy of the model trained on the remaining points to predict backdoored examples as the target label. A successful data removal would yield a lower attack accuracy. Figure 3 I.(b) shows that our method already takes effect in the early stages, whereas other baselines can start defending from the attack only after removing over $13,000$ samples. The efficacy of LAVA is in part attributable to inspection of distances between both features and labels. The backdoored training samples that are poisoned to the target class will be

"unnatural" in that class, i.e., they have a large feature distance from the original samples in the target class. While the poisoned examples contain a small feature perturbation compared to the natural examples from some other classes, their label distance to them is large because their labels are altered.

**Poisoning Attack Detection.** Poisoning attacks are similar to backdoor attacks in the sense that they both inject adversarial points into the training set to manipulate the prediction of certain test examples. However, poisoning attacks are considered unable to control test examples. We consider a popular attack termed "feature-collision" attack (Shafahi et al., 2018), where we select a target sample from the *Cat* class test set and blend the selected image with the chosen target class training samples, *Frog* in our case. In this attack, we do not modify labels and blend the *Cat* image only into 50 (0.1%) samples of *Frog*, which makes this attack especially hard to detect. During inference time, we expect the attacked model to consistently classify the chosen *Cat* as a *Frog*. In Figure 3 II.(a), we observe that LAVA outperforms all baselines and achieves an 80% detection rate by removing only 11k samples, which is around 60% fewer samples than the highest baseline. Figure 3 II.(b) shows that by removing data according to LAVA ranking, the target model has reduced the confidence of predicting the target *Cat* sample as a *Frog* to below 40%. Our technique leverages the fact that the features from a different class are mixed with the features of the poisoned class, which increases the feature distance between the poisoned and non-poisoned *Frog* examples.

**Noisy Feature Detection.** While adding small Gaussian noises to training samples may benefit model robustness (Rusak et al., 2020), strong noise, such as due to sensor failure, can significantly affect the model performance. We add strong white noise to 25% of all CIFAR-10 dataset without changing any labels. Our method performs extremely well as shown in Figure 3 III.(a) and detects all 12,500 noisy samples by inspecting less than 15,000 samples. This explains the sudden drop of the model's accuracy at the removal budget of 15,000 samples in Figure 3 III.(b): the model starts throwing away only clean samples from that point. LAVA performs well in this scenario since the strong noise increases the feature distance significantly.

**Mislabeled Data Detection.** Due to the prevalence of human labeling errors (Karimi et al., 2020), it is crucial to detect mislabeled samples. We shuffle labels of 25% samples in the CIFAR-10 dataset to random classes. Unlike backdoor and poisoning attacks, this case is especially harder to detect since wrong samples are spread out throughout classes instead of all placed inside a target class. However, as shown in Figure 3 IV.(a), LAVA's detection rate outperforms other baselines and the model performance is maintained even after 20k of removed data (Figure IV.(b)).

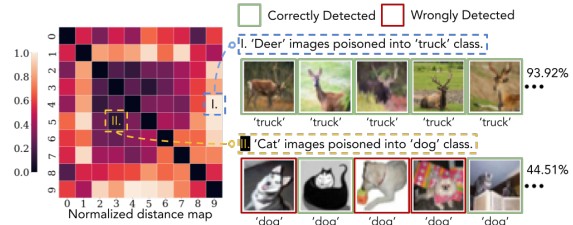

Figure 4: Left: Heatmap of inter-class distance of CIFAR-10. Right: Examples of irrelevant data. The detection rates of first 500 inspected images are 94% and 46% for *Deer-Truck*, *Cat-Dog*, respectively.

**Irrelevant Data Detection.** Often the collected datasets through web scraping have irrelevant samples in given classes (Northcutt et al., 2021; Tsipras et al., 2020), e.g., in a class of *Glasses*, we might have both water glass and eyeglasses due to lack of proper inspection or class meaning specification. This case is different from the mislabeled data scenario, in which case the training features are all relevant to the task. Since the irrelevant examples are highly likely to have completely different features than the desired class representation, LAVA is expected to detect these examples. We design an experiment where we remove all images of one specific class from the classification output but split them equally to the other remaining classes as irrelevant images. As shown in Figure 4, the detection result over a class varies based on the distance between that class and the class from which irrelevant images are drawn. For instance, when *Deer* images are placed into the *Truck* class, we can detect almost 94% of all *Deer* images within first 500 removed images. On the other hand, when we place *Cat* images into *dog* class, our detection rate drops to 45% within the top 500.

**Computational Efficiency.** So far, we have focused on the method's performance without considering the actual runtime. We compare the runtime-performance tradeoff on the CIFAR-10 example of 2000 samples with 10% backdoor data, a scale in which every baseline can be executed in a reasonable time. As shown in Figure 5, our method achieves a significant improvement in efficiency while being able to detect bad data more effectively.

**Dependence on Validation Data Size.** For current experiments, we have assumed the validation set of size $10K$. Such a scale of data is not hard to acquire, as one can get high-quality data from crowdsourcing platforms, such as Amazon Mechanical Turk for \$12 per each $1K$ samples (AWS, 2019). While our method achieves remarkable performance when using $10K$ validation data, we perform ablation study on much smaller sets (Appendix B.2.1), where LAVA, notably, can still outperform other baselines. As an example on mislabeled data detection, our method with $2K$ validation data achieves $80\%$ detection rate at data removal budget of $25K$ (Fig. 9), whereas the best performing baseline achieves such a performance with 5 times bigger validation data, $10K$ (Fig. 3 IV.(a)). Furthermore, even on a tiny validation set of size $500$, LAVA consistently outperforms all the baselines with the same validation size (Fig. 11). This shows that our method remains effective performance for various sizes of validation data.

## 5 RELATED WORK

Existing data valuation methods include LOO and influence function (Koh & Liang, 2017), the Shapley value (Jia et al., 2019b; Ghorbani & Zou, 2019; Wang & Jia, 2023), the Banzhaf value (Wang & Jia, 2022), Least Cores (Yan & Procaccia, 2021), Beta Shapley (Kwon & Zou, 2021), and reinforcement learning-based method (Yoon et al., 2020). However, they all assume the knowledge of the underlying learning algorithms and suffer large computational complexity. The work of Jia et al. (2019a) has proposed to use $K$-Nearest Neighbor Classifier as a default proxy model to perform data valuation. While it can be thought of as a learning-

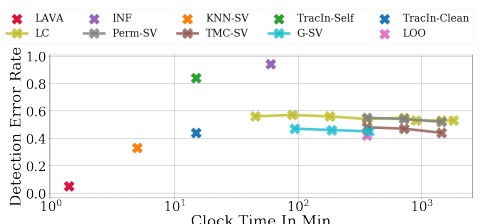

Figure 5: Runtime v.s. Detection Error comparison between LAVA and baselines on inspecting 2000 samples from CIFAR-10 with $10\%$ backdoor data.

agnostic data valuation method, it is not as effective and efficient as our method in distinguishing data quality. Xu et al. (2021) propose to use the volume to measure the utility of a dataset. Volume is agnostic to learning algorithms and easy to calculate because is defined simply as the square root of the trace of feature matrix inner product. However, the sole dependence on features makes it incapable of detecting bad data caused by labeling errors. Moreover, to evaluate the contribution of individual points, the authors propose to resort to the Shapley value, which would still be expensive for large datasets.

## 6 DISCUSSION AND OUTLOOK

This paper describes a learning-agnostic data valuation framework. In particular, in contrast to existing methods which typically adopt model validation performance as the utility function, we approximate the utility of a dataset based on its class-wise Wasserstein distance to a given validation set and provide theoretical justification for this approximation. Furthermore, we propose to use the calibrated gradients of the OT distance to value individual datapoints, which can be obtained for free if one uses an off-the-shelf solver to calculate the Wasserstein distance. Importantly, we have tested on various datasets, and our LAVA framework can significantly improve the state-of-the-art performance of using data valuation methods to detect bad data while being substantially more efficient. Due to the stochasticity of ML and the inherent tolerance to noise, it is often challenging to identify low-quality data by inspecting their influence on model performance scores. The take-away from our empirical study is that despite being extensively adopted in the past, low-quality data detection through model performance changes is actually *suboptimal*; lifting the dependence of data valuation on the actual learning process provides a better pathway to distinguish data quality.

Despite the performance and efficiency improvement, our work still has some limitations. As a result, it opens up many new investigation venues: **(1)** How to further lift the dependence on validation data? While a validation set representative of the downstream learning task is a common assumption in the ML literature, it may or may not be available during data exchange. **(2)** Our design could be vulnerable to existing poisons that directly or indirectly minimize the similarity to clean data (Huang et al., 2021; Pan et al., 2022). Further investigation into robust data valuation would be intriguing. **(3)** Our current method does not have enough flexibility for tasks that aim for goals beyond accuracy, e.g., fairness. Folding other learning goals in is an exciting direction. **(4)** Customizing the framework to natural language data is also of practical interest.

## 7 ACKNOWLEDGEMENTS

RJ and the ReDS Lab gratefully acknowledge the support from the Cisco Research Award, the Virginia Tech COE Fellowship, and the NSF CAREER Award. Jiachen T. Wang is supported by Princeton's Gordon Y. S. Wu Fellowship. YZ is supported by the Amazon Fellowship.

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

## APPENDIX A    RESTATEMENT OF THEOREMS AND FULL PROOFS

In this section, we will restate our main results and give full proofs.

### A.1    SUMMARY OF NOTATIONS

Let $\mu_t$, $\mu_v$ be the training distribution and validation distribution, respectively. We denote $f_t : \mathcal{X} \to \{0,1\}^V$, $f_v : \mathcal{X} \to \{0,1\}^V$ as the labeling functions for training and validation data, where $V$ is the number of different labels. We can then denote the ***joint distribution of random data-label pairs*** $(x, f_t(x))_{x \sim \mu_t(x)}$ and $(x, f_v(x))_{x \sim \mu_v(x)}$ ***as*** $\mu_t^{f_t}$ ***and*** $\mu_v^{f_v}$***, respectively, which are the same notations as*** $\mu_t$ ***and*** $\mu_v$ ***but made with explicit dependence on*** $f_t$ ***and*** $f_v$ ***for clarity***. The distributions of $(f_t(x))_{x \sim \mu_t(x)}$, $(f_v(x))_{x \sim \mu_v(x)}$ are denoted as $\mu_{f_t}, \mu_{f_v}$, respectively. Besides, we define conditional distributions $\mu_t(x|y) := \frac{\mu_t(x)I[f_t(x)=y]}{\int \mu_t(x)I[f_t(x)=y]dx}$ and $\mu_v(x|y) := \frac{\mu_v(x)I[f_v(x)=y]}{\int \mu_v(x)I[f_v(x)=y]dx}$. Let $f : \mathcal{X} \to [0,1]^V$ be the model trained on training data and $\mathcal{L} : \{0,1\}^V \times [0,1]^V \to \mathbb{R}^+$ be the loss function. We denote $\pi \in \Pi(\mu_1, \mu_2)$ as a coupling between a pair of distributions $\mu_1, \mu_2$ and $\mathrm{d} : \mathcal{X} \times \mathcal{X} \to \mathbb{R}$ as a distance metric function.

The 1-Wasserstein distance with respect to distance function $\mathrm{d}$ between two distributions $\mu_1, \mu_2$ is defined as $W_{\mathrm{d}}(\mu_1, \mu_2) := \inf_{\pi \in \Pi(\mu_1,\mu_2)} \mathbb{E}_{(x,y) \sim \pi} [\mathrm{d}(x,y)]$. More generally, the 1-Wasserstein distance with respect to cost function $\mathcal{C}$ is defined as $W_{\mathcal{C}}(\mu_1, \mu_2) := \inf_{\pi \in \Pi(\mu_1,\mu_2)} \mathbb{E}_{(x,y) \sim \pi} [\mathcal{C}(x,y)]$.

### A.2    STATEMENT OF ASSUMPTIONS

To prove Theorem 1, we need the concept of probabilistic *cross*-Lipschitzness, which assumes that two labeling functions should produce consistent labels with high probability on two close instances.

**Definition 3** (Probabilistic Cross-Lipschitzness). *Two labeling functions* $f_t : \mathcal{X} \to \{0,1\}^V$ *and* $f_v : \mathcal{X} \to \{0,1\}^V$ *are* $(\epsilon, \delta)$*-probabilistic cross-Lipschitz w.r.t. a joint distribution* $\pi$ *over* $\mathcal{X} \times \mathcal{X}$ *if for all* $\epsilon > 0$:

$$P_{(x_1,x_2) \sim \pi}[\|f_t(x_1) - f_v(x_2)\| > \epsilon \mathrm{d}(x_1, x_2)] \le \delta. \tag{4}$$

Intuitively, given labeling functions $f_t$, $f_v$ and a coupling $\pi$, we can bound the probability of finding pairs of training and validation instances labelled differently in a $(1/\epsilon)$-ball with respect to $\pi$.

**Our Assumptions.**    Assuming that $f$ is an $\epsilon$-Lipschitz function. Given a metric function $\mathrm{d}(\cdot, \cdot)$, we define a cost function $\mathcal{C}$ between $(x_t, y_t)$ and $(x_v, y_v)$ as

$$\mathcal{C}((x_t, y_t), (x_v, y_v)) := \mathrm{d}(x_t, x_v) + cW_{\mathrm{d}}(\mu_t(\cdot|y_t), \mu_v(\cdot|y_v)), \tag{5}$$

where $c$ is a constant. Let $\pi_{x,y}^*$ be the coupling between $\mu_t^{f_t}, \mu_v^{f_v}$ such that

$$\pi_{x,y}^* := \underset{\pi \in \Pi(\mu_t^{f_t}, \mu_v^{f_v})}{\arg \inf} \mathbb{E}_{((x_t,y_t),(x_v,y_v)) \sim \pi}[\mathcal{C}((x_t, y_t), (x_v, y_v))]. \tag{6}$$

We define two couplings $\pi^*$ and $\widetilde{\pi}^*$ between $\mu_t(x), \mu_v(x)$ as follows:

$$\pi^*(x_t, x_v) := \int_{\mathcal{Y}} \int_{\mathcal{Y}} \pi_{x,y}^*((x_t, y_t), (x_v, y_v)) \, dy_t dy_v. \tag{7}$$

For $\widetilde{\pi}^*$, we first need to define a coupling between $\mu_{f_t}, \mu_{f_v}$:

$$\pi_y^*(y_t, y_v) := \int_{\mathcal{X}} \int_{\mathcal{X}} \pi_{x,y}^*((x_t, y_t), (x_v, y_v)) \, dx_t dx_v \tag{8}$$

and another coupling between $\mu_t^{f_t}, \mu_v^{f_v}$:

$$\widetilde{\pi}_{x,y}^*((x_t, y_t), (x_v, y_v)) := \pi_y^*(y_t, y_v)\mu_t(x_t|y_t)\mu_v(x_v|y_v). \tag{9}$$

Finally, $\widetilde{\pi}^*$ is constructed as follows:

$$\widetilde{\pi}^*(x_t, x_v) := \int_{\mathcal{Y}} \int_{\mathcal{Y}} \pi_y^*(y_t, y_v)\mu_t(x_t|y_t)\mu_v(x_v|y_v)\, dy_t dy_v. \tag{10}$$

It is easy to see that all joint distributions defined above are couplings between the corresponding distribution pairs.

We assume that $f_t$, $f_v$ are $(\epsilon_{tv}, \delta_{tv})$-probabilistic cross-Lipschitz with respect to $\widetilde{\pi}^*$ in metric d. Additionally, we assume that $\epsilon_{tv}/\epsilon \leq c$ and the loss function $\mathcal{L}$ is $k$-Lipschitz in both inputs. Besides, from their definitions above, we have that $\|f(x)\|, \|f_t(x)\|, \|f_v(x)\| \leq V$.

The assumption of probabilistic cross-Lipschitzness would be violated only when the underlying coupling assigns large probability to pairs of training-validation features that are close enough (within $1/\epsilon_{tv}$-ball) but labeled differently. However, $\widetilde{\pi}^*$ is generally not such a coupling. Note that $\pi^*$ is the optimal coupling between training and validation distributions that minimizes a cost function $\mathcal{C}$ pertaining to both feature and label space. Hence, $\pi_y^*(y_t, y_v)$, the marginal distribution of $\pi^*$ over the training and validation label space, tends to assign high probability to those label pairs that agree. On the other hand, $\widetilde{\pi}_{x,y}^*$ can be thought of as a coupling that first generates training-validation labels from $\pi_y^*$ and then generates the features in each dataset conditioning on the corresponding labels. Hence, the marginal distribution $\widetilde{\pi}^*$ of training-validation feature pairs generated by $\widetilde{\pi}_{x,y}^*$ would assign high likelihood to those features with the same labels. So, conceptually, the probabilistic cross-Lipschitzness assumption should be easily satisfied by $\widetilde{\pi}^*$.

## A.3 DETAILED PROOF

**Theorem 1** (restated). *Given the above assumptions, we have*

$$\mathbb{E}_{x \sim \mu_v(x)}\left[\mathcal{L}(f_v(x), f(x))\right] \leq \mathbb{E}_{x \sim \mu_t(x)}\left[\mathcal{L}(f_t(x), f(x))\right] + k\epsilon W_{\mathcal{C}}(\mu_t^{f_t}, \mu_v^{f_v}) + 2kV\delta_{tv}. \tag{11}$$

*Proof.*

$$\mathbb{E}_{x \sim \mu_v(x)}[\mathcal{L}(f_v(x), f(x))] \tag{12}$$
$$= \mathbb{E}_{x \sim \mu_v(x)}[\mathcal{L}(f_v(x), f(x))] - \mathbb{E}_{x \sim \mu_t(x)}[\mathcal{L}(f_t(x), f(x))] + \mathbb{E}_{x \sim \mu_t(x)}[\mathcal{L}(f_t(x), f(x))] \tag{13}$$
$$\leq \mathbb{E}_{x \sim \mu_t(x)}[\mathcal{L}(f_t(x), f(x))] + \left|\mathbb{E}_{x \sim \mu_v(x)}[\mathcal{L}(f_v(x), f(x))] - \mathbb{E}_{x \sim \mu_t(x)}[\mathcal{L}(f_t(x), f(x))]\right|. \tag{14}$$

We bound $\left|\mathbb{E}_{x \sim \mu_v(x)}\left[\mathcal{L}(f_v(x), f(x))\right] - \mathbb{E}_{x \sim \mu_t(x)}\left[\mathcal{L}(f_t(x), f(x))\right]\right|$ as follows:

$$\left|\mathbb{E}_{x \sim \mu_v(x)}\left[\mathcal{L}(f_v(x), f(x))\right] - \mathbb{E}_{x \sim \mu_t(x)}\left[\mathcal{L}(f_t(x), f(x))\right]\right| \tag{15}$$

$$= \left|\int_{\mathcal{X}^2} [\mathcal{L}(f_v(x_v), f(x_v)) - \mathcal{L}(f_t(x_t), f(x_t))]\, d\pi^*(x_t, x_v)\right| \tag{16}$$

$$= \left|\int_{\mathcal{X}^2} [\mathcal{L}(f_v(x_v), f(x_v)) - \mathcal{L}(f_v(x_v), f(x_t)) + \mathcal{L}(f_v(x_v), f(x_t)) - \mathcal{L}(f_t(x_t), f(x_t))]\, d\pi^*(x_t, x_v)\right| \tag{17}$$

$$\leq \underbrace{\int_{\mathcal{X}^2} |\mathcal{L}(f_v(x_v), f(x_v)) - \mathcal{L}(f_v(x_v), f(x_t))|\, d\pi^*(x_t, x_v)}_{U_1} \tag{18}$$

$$+ \underbrace{\int_{\mathcal{X}^2} |\mathcal{L}(f_v(x_v), f(x_t)) - \mathcal{L}(f_t(x_t), f(x_t))|\, d\pi^*(x_t, x_v)}_{U_2}, \tag{19}$$

where the last inequality is due to triangle inequality.

Now, we bound $U_1$ and $U_2$ separately. For $U_1$, we have

$$U_1 \leq k \int_{\mathcal{X}^2} \|f(x_v) - f(x_t)\| \, d\pi^*(x_t, x_v) \tag{20}$$

$$\leq k\epsilon \int_{\mathcal{X}^2} \mathrm{d}(x_t, x_v) \, d\pi^*(x_t, x_v), \tag{21}$$

where both inequalities are due to Lipschitzness of $\mathcal{L}$ and $f$.

In order to bound $U_2$, we first recall that $\pi_y^*(y_t, y_v) = \int_{\mathcal{X}} \int_{\mathcal{X}} \pi_{x,y}^*((x_t, y_t), (x_v, y_v)) \, dx_t dx_v$ and $\widetilde{\pi}_{x,y}^*((x_t, y_t), (x_v, y_v)) := \pi_y^*(y_t, y_v) \mu_t(x_t|y_t) \mu_v(x_v|y_v)$:

Observe that

$$U_2 = \int_{\mathcal{X}^2} \int_{\mathcal{Y}^2} |\mathcal{L}(f_v(x_v), f(x_t)) - \mathcal{L}(f_t(x_t), f(x_t))| \, d\pi_{x,y}^*((x_t, y_t), (x_v, y_v)) \tag{22}$$

$$= \int_{\mathcal{Y}^2} \int_{\mathcal{X}^2} |\mathcal{L}(y_v, f(x_t)) - \mathcal{L}(y_t, f(x_t))| \, d\pi_{x,y}^*((x_t, y_t), (x_v, y_v)) \tag{23}$$

$$\leq k \int_{\mathcal{Y}^2} \int_{\mathcal{X}^2} \|y_v - y_t\| \, d\pi_{x,y}^*((x_t, y_t), (x_v, y_v)) \tag{24}$$

$$= k \int_{\mathcal{Y}^2} \|y_v - y_t\| \, d\pi_y^*(y_t, y_v), \tag{25}$$

where the second equality is due to a condition that if $y_t \neq f_t(x_t)$ or $y_v \neq f_v(x_v)$, then $\pi_{x,y}^*((x_t, y_t), (x_v, y_v)) = 0$.

Now we can bound $U_2$ as follows:

$$U_2 \leq k \int_{\mathcal{Y}^2} \|y_v - y_t\| \, d\pi_y^*(y_t, y_v) \tag{26}$$

$$= k \int_{\mathcal{X}^2} \int_{\mathcal{Y}^2} \|y_v - y_t\| \, d\widetilde{\pi}_{x,y}^*((x_t, y_t), (x_v, y_v)) \tag{27}$$

$$= k \int_{\mathcal{Y}^2} \int_{\mathcal{X}^2} \|f_v(x_v) - f_t(x_t)\| \, d\widetilde{\pi}_{x,y}^*((x_t, y_t), (x_v, y_v)), \tag{28}$$

where the last step holds since if $y_t \neq f_t(x_t)$ or $y_v \neq f_v(x_v)$ then $\widetilde{\pi}_{x,y}^*((x_t, y_t), (x_v, y_v)) = 0$.

Define the region $A = \{(x_t, x_v) : \|f_v(x_v) - f_t(x_t)\| < \epsilon_{tv} \mathrm{d}(x_t, x_v)\}$, then

$$k \int_{\mathcal{Y}^2} \int_{\mathcal{X}^2} \|f_v(x_v) - f_t(x_t)\| \, d\widetilde{\pi}_{x,y}^*((x_t, y_t), (x_v, y_v)) \tag{29}$$

$$= k \int_{\mathcal{Y}^2} \int_{\mathcal{X}^2 \backslash A} \|f_v(x_v) - f_t(x_t)\| \, d\widetilde{\pi}_{x,y}^*((x_t, y_t), (x_v, y_v)) \tag{30}$$

$$+ k \int_{\mathcal{Y}^2} \int_A \|f_v(x_v) - f_t(x_t)\| \, d\widetilde{\pi}_{x,y}^*((x_t, y_t), (x_v, y_v)) \tag{31}$$

$$\leq k \int_{\mathcal{Y}^2} \int_{\mathcal{X}^2 \backslash A} 2V \, d\widetilde{\pi}_{x,y}^*((x_t, y_t), (x_v, y_v)) \tag{32}$$

$$+ k \int_{\mathcal{Y}^2} \int_A \|f_v(x_v) - f_t(x_t)\| \, d\widetilde{\pi}_{x,y}^*((x_t, y_t), (x_v, y_v)). \tag{33}$$

Let's define $\tilde{f}_t(x_t) = f_t(x_t)$ and $\tilde{f}_v(x_v) = f_v(x_v)$ if $(x_t, x_v) \in A$, and $\tilde{f}_t(x_t) = \tilde{f}_v(x_v) = 0$ otherwise (note that $\|\tilde{f}_v(x_v) - \tilde{f}_t(x_t)\| \leq \epsilon_{tv} \mathrm{d}(x_t, x_v)$ for all $(x_t, x_v) \in \mathcal{X}^2$), then we can bound the second term as follows:

$$k \int_{\mathcal{Y}^2} \int_A \|f_v(x_v) - f_t(x_t)\| \, d\widetilde{\pi}_{x,y}^*((x_t, y_t), (x_v, y_v)) \tag{34}$$

$$\leq k \int_{\mathcal{Y}^2} d\pi_y^*(y_t, y_v) \int_A \|f_v(x_v) - f_t(x_t)\| \, d\mu_t(x_t|y_t) d\mu_v(x_v|y_v) \tag{35}$$

$$= k \int_{\mathcal{Y}^2} d\pi_y^*(y_t, y_v) \int_{\mathcal{X}^2} \left\| \tilde{f}_v(x_v) - \tilde{f}_t(x_t) \right\| \, d\mu_t(x_t|y_t) d\mu_v(x_v|y_v) \tag{36}$$

$$= k \int_{\mathcal{Y}^2} d\pi_y^*(y_t, y_v) \left\| \mathbb{E}_{x_v \sim \mu_v(\cdot|y_v)}[\tilde{f}_v(x_v)] - \mathbb{E}_{x_t \sim \mu_v(\cdot|y_t)}[\tilde{f}_t(x_t)] \right\| \tag{37}$$

$$\leq k\epsilon_{tv} \int_{\mathcal{Y}^2} d\pi_y^*(y_t, y_v) W_{\mathrm{d}}(\mu_t(\cdot|y_t), \mu_v(\cdot|y_v)). \tag{38}$$

Inequality (38) is a consequence of the duality form of the Kantorovich-Rubinstein theorem (Villani (2021), Chapter 1).

Combining two parts, we have

$$U_2 \leq k \int_{\mathcal{Y}^2} \int_{\mathcal{X}^2 \backslash A} 2V \, d\widetilde{\pi}_{x,y}^*((x_t, y_t), (x_v, y_v)) \tag{39}$$

$$+ k\delta_{tv} \int_{\mathcal{Y}^2} d\pi_y^*(y_t, y_v) W_{\mathrm{d}}(\mu_t(\cdot|y_t), \mu_v(\cdot|y_v)) \tag{40}$$

$$\leq 2kV\delta_{tv} + k\epsilon_{tv} \int_{\mathcal{Y}^2} d\pi_y^*(y_t, y_v) W_{\mathrm{d}}(\mu_t(\cdot|y_t), \mu_v(\cdot|y_v)), \tag{41}$$

where the last step is due to the probabilistic cross-Lipschitzness of $f_t, f_v$ with respect to $\widetilde{\pi}_{x,y}^*$.

Now, combining the bound for $U_1$ and $U_2$, we have

$$\mathbb{E}_{x \sim \mu_v(x)}[\mathcal{L}(f_v(x), f(x))] - \mathbb{E}_{x \sim \mu_t(x)}[\mathcal{L}(f_t(x), f(x))] \tag{42}$$

$$\leq k\epsilon \int_{\mathcal{X}^2} \mathrm{d}(x_t, x_v) d\pi(x_t, x_v) + 2kV\delta_{tv} + k\epsilon_{tv} \int_{\mathcal{Y}^2} d\pi_y^*(y_t, y_v) W_{\mathrm{d}}(\mu_t(\cdot|y_t), \mu_v(\cdot|y_v)) \tag{43}$$

$$= k \int_{(\mathcal{X} \times \mathcal{Y})^2} [\epsilon \mathrm{d}(x_t, x_v) + \epsilon_{tv} W_{\mathrm{d}}(\mu_t(\cdot|y_t), \mu_v(\cdot|y_v))] \, d\pi_{x,y}^*((x_t, y_t), (x_v, y_v)) + 2kV\delta_{tv} \tag{44}$$

$$\leq k \int_{(\mathcal{X} \times \mathcal{Y})^2} [\epsilon \mathrm{d}(x_t, x_v) + c\epsilon W_{\mathrm{d}}(\mu_t(\cdot|y_t), \mu_v(\cdot|y_v))] \, d\pi_{x,y}^*((x_t, y_t), (x_v, y_v)) + 2kV\delta_{tv} \tag{45}$$

$$= k\epsilon \mathbb{E}_{\pi_{x,y}^*}[\mathcal{C}((x_t, y_t), (x_v, y_v))] + 2kV\delta_{tv} \tag{46}$$

$$= k\epsilon W_{\mathcal{C}}(\mu_t^{f_t}, \mu_v^{f_v}) + 2kV\delta_{tv}, \tag{47}$$

where the last step is due to the definition of $\pi_{x,y}^*$. This leads to the final conclusion. $\qquad\square$

**Theorem 5** (restated). *Let* $\mathrm{OT}(\mu_t, \mu_v)$ *and* $\mathrm{OT}_\varepsilon(\mu_t, \mu_v)$ *be the original formulation and entropy penalized formulation (as defined in Subsection 2.1) for the OT problem between the empirical measures* $\mu_t$ *and* $\mu_v$ *associated with the two data sets* $D_t$ *and* $D_v$, *respectively. Then, for any* $i \neq j \neq k \in \{1, 2, ...N\}$ *and* $o \neq p \neq q \in \{1, 2, ...M\}$, *the difference between the calibrated gradients for two datapoints* $z_i$ *and* $z_k$ *in dataset* $D_t$ *and the difference for* $z_p'$ *and* $z_q'$ *in* $D_v$ *can be calculated as*

$$\frac{\partial \mathrm{OT}(\mu_t, \mu_v)}{\partial \mu_{\mathrm{t}}(z_i)} - \frac{\partial \mathrm{OT}(\mu_t, \mu_v)}{\partial \mu_{\mathrm{t}}(z_k)} = \frac{\partial \mathrm{OT}_\varepsilon(\mu_t, \mu_v)}{\partial \mu_{\mathrm{t}}(z_i)} - \frac{\partial \mathrm{OT}_\varepsilon(\mu_t, \mu_v)}{\partial \mu_{\mathrm{t}}(z_k)} - \varepsilon \cdot \frac{N}{N-1} \cdot \left( \frac{1}{(\pi_\varepsilon^*)_{kj}} - \frac{1}{(\pi_\varepsilon^*)_{ij}} \right),$$

$$\frac{\partial \mathrm{OT}(\mu_t, \mu_v)}{\partial \mu_{\mathrm{v}}(z_p')} - \frac{\partial \mathrm{OT}(\mu_t, \mu_v)}{\partial \mu_{\mathrm{v}}(z_q')} = \frac{\partial \mathrm{OT}_\varepsilon(\mu_t, \mu_v)}{\partial \mu_{\mathrm{v}}(z_p')} - \frac{\partial \mathrm{OT}_\varepsilon(\mu_t, \mu_v)}{\partial \mu_{\mathrm{v}}(z_q')} - \varepsilon \cdot \frac{M}{M-1} \cdot \left( \frac{1}{(\pi_\varepsilon^*)_{oq}} - \frac{1}{(\pi_\varepsilon^*)_{op}} \right),$$

*where* $\pi_\varepsilon^*$ *is the optimal primal solution to the entropy penalized OT problem,* $z_j$ *is any datapoint in* $D_t$ *other than* $z_i$ *or* $z_k$, $z_o'$ *is any datapoint in* $D_v$ *other than* $z_p'$ *or* $z_q'$, $|D_t| = N$, *and* $|D_v| = M$.

*Proof.* Let $\mathcal{L}(\pi, f, g)$ and $\mathcal{L}_\varepsilon(\pi_\varepsilon, f_\varepsilon, g_\varepsilon)$ be the Lagrangian functions for original formulation and entropy penalized formulation between the datasets $D_t$ and $D_v$, respectively, which can be written as

$$\mathcal{L}(\pi, f, g) = \langle \pi, c \rangle + \sum_{i=1}^N f_i \cdot (\pi'_i \cdot I_N - a_i) + \sum_{j=1}^M g_j \cdot (I'_M \cdot \pi_j - b_j),$$

$$\mathcal{L}_\varepsilon(\pi_\varepsilon, f_\varepsilon, g_\varepsilon) = \langle \pi_\varepsilon, c \rangle + \varepsilon \cdot \sum_{i=1}^N \sum_{j=1}^M \log \frac{(\pi_\varepsilon)_{ij}}{\mu_t(z_i) \cdot \mu_v(z_j)} + \sum_{i=1}^N (f_\varepsilon)_i \cdot [(\pi_\varepsilon)'_i \cdot I_M - \mu_t(z_i)]$$

$$+ \sum_{j=1}^M (g_\varepsilon)_j \cdot [I'_N \cdot (\pi_\varepsilon)_j - \mu_v(z_j)],$$

where $c^{N \times M}$ is the cost matrix consisting of distances between $N$ datapoints in $D_t$ and $M$ datapoints in $D_v$, $I_N = (1, 1, ...1) \in \mathbb{R}^{N \times 1}$ and $I'_M = (1, 1, ...1)^T \in \mathbb{R}^{1 \times M}$, $\pi$ and $(f, g)$ denote the primal and dual variables, and $\pi'_i$ and $\pi_j$ denote the $i^{th}$ row and $j^{th}$ column in matrix $\pi$, respectively.

The first-order necessary condition for optima in Lagrangian Multiplier Theorem

gives that

$$\nabla \mathcal{L}_\pi(\pi^*, f^*, g^*) = 0 \quad \text{and} \quad \nabla(\mathcal{L}_\varepsilon)_\pi((\pi_\varepsilon)^*, (f_\varepsilon)^*, (g_\varepsilon)^*) = 0,$$

where $\pi^*$ and $(f^*, g^*)$ denote the optimal solutions to the primal and dual problems, respectively. Thus, for any $i \in \{1, 2, \ldots, N\}$ and $j \in \{1, 2, \ldots, M\}$, we have

$$\nabla \mathcal{L}_\pi(\pi^*, f^*, g^*)_{ij} = c_{ij} + f_i^* + g_j^* = 0,$$

$$\nabla(\mathcal{L}_\varepsilon)_\pi(\pi_\varepsilon^*, f_\varepsilon^*, g_\varepsilon^*)_{ij} = c_{ij} + \varepsilon \cdot \frac{1}{(\pi_\varepsilon^*)_{ij}} + (f_\varepsilon)_i^* + (g_\varepsilon)_j^* = 0.$$

Subtracting, we have

$$[f_i^* - (f_\varepsilon)_i^*] + [g_j^* - (g_\varepsilon)_j^*] - \varepsilon \cdot \frac{1}{(\pi_\varepsilon^*)_{ij}} = 0.$$

Then, for $\forall k \neq i \in \{1, 2, ...N\}$, we have

$$[f_k^* - (f_\varepsilon)_k^*] + [g_j^* - (g_\varepsilon)_j^*] - \varepsilon \cdot \frac{1}{(\pi_\varepsilon^*)_{kj}} = 0.$$

Subtracting and reorganizing, we get

$$[(f_\varepsilon)_i^* - (f_\varepsilon)_k^*] = (f_i^* - f_k^*) - \varepsilon \cdot \left[ \frac{1}{(\pi_\varepsilon^*)_{ij}} - \frac{1}{(\pi_\varepsilon^*)_{kj}} \right].$$

From the definition of the calibrated gradients in Eq.1, we have

$$\frac{\partial \operatorname{OT}(\mu_t, \mu_v)}{\partial \mu_t(z_i)} - \frac{\partial \operatorname{OT}(\mu_t, \mu_v)}{\partial \mu_t(z_k)} = \frac{N}{N-1} (f_i^* - f_k^*),$$

$$\frac{\partial \operatorname{OT}_\varepsilon(\mu_t, \mu_v)}{\partial \mu_t(z_i)} - \frac{\partial \operatorname{OT}_\varepsilon(\mu_t, \mu_v)}{\partial \mu_t(z_k)} = \frac{N}{N-1} [(f_\varepsilon)_i^* - (f_\varepsilon)_k^*].$$

Finally, subtracting and reorganizing, we have

$$\frac{\partial \operatorname{OT}_\varepsilon(\mu_t, \mu_v)}{\partial \mu_t(z_i)} - \frac{\partial \operatorname{OT}_\varepsilon(\mu_t, \mu_v)}{\partial \mu_t(z_k)} = \frac{\partial \operatorname{OT}(\mu_t, \mu_v)}{\partial \mu_t(z_i)} - \frac{\partial \operatorname{OT}(\mu_t, \mu_v)}{\partial \mu_t(z_k)} - \varepsilon \cdot \frac{N}{N-1} \cdot \left[ \frac{1}{(\pi_\varepsilon^*)_{ij}} - \frac{1}{(\pi_\varepsilon^*)_{kj}} \right].$$

The proof for the second part of the Theorem is similar.

$$\frac{\partial \operatorname{OT}_\varepsilon(\mu_t, \mu_v)}{\partial \mu_v(z'_p)} - \frac{\partial \operatorname{OT}_\varepsilon(\mu_t, \mu_v)}{\partial \mu_v(z'_q)} = \frac{\partial \operatorname{OT}(\mu_t, \mu_v)}{\partial \mu_v(z'_p)} - \frac{\partial \operatorname{OT}(\mu_t, \mu_v)}{\partial \mu_v(z'_q)} - \varepsilon \cdot \frac{M}{M-1} \cdot \left[ \frac{1}{(\pi_\varepsilon^*)_{op}} - \frac{1}{(\pi_\varepsilon^*)_{oq}} \right].$$

Then the proof is complete. $\qquad \square$

# APPENDIX B  ADDITIONAL EXPERIMENTAL RESULTS

## B.1  EVALUATING DATA VALUATION USE CASES ON DIVERSE DATASETS

In the main text, we have focused our evaluation on CIFAR-10. Here, we provide experiments to show effectiveness of LAVA on diverse datasets for detecting bad data.

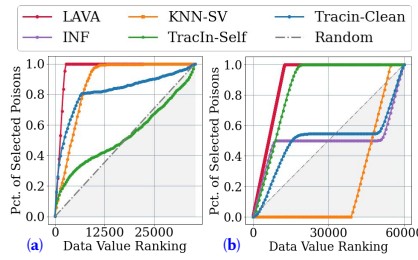

**Backdoor Attack Detection.** We evaluate another type of backdoor attack (Section 4), which is the Hello Kitty blending attack (Blend) (Chen et al., 2017) that mixes the target class sample with the Hello Kitty image, as illustrated in Figure 8 (B). We attack the German Traffic Sign dataset (GTSRB) on the target class 6 by poisoning 1764 (5%) samples of the whole dataset. Our method achieves the highest detection rate, as shown in Figure 6(a). In particular, the 5000 points with lowest data values contain all poisoned data based on the LAVA data values, while the second best method on this task, KNN-SV, can cover all poisoned examples with around 11,000 samples. Our algorithm performs especially well for this attack, since the label of poisoned data is changed to the target class and the patching trigger is large. Both the label and feature changes contribute to the increase of the OT distance and thus ease the detection.

Figure 6: (a) Blend backdoor attack detection on GTSRB dataset. Comparison with baselines. (b) Noisy feature detection on the digit MNIST dataset. Comparison with baselines.



**Noisy Feature Detection.** Here, we show the usage of LAVA on the MNIST dataset where 25% of the whole dataset is contaminated by feature noise. Our method still outperforms all the baselines by detecting all noisy data within first 14,000 samples, which is 5,000 less than the best baseline would require, which is shown in Figure 6(b).

**Irrelevant Data.** We perform another irrelevant data detection experiment and focus on the CIFAR100 dataset. In Figure 7, we illustrate some of the irrelevant samples detected by LAVA. Intuitively, irrelevant data in the class should be easily detected by LAVA, since the images are far from the representative of the class and increasing the probability mass associated with these images leads to larger distributional distance to the clean validation data.

Figure 7: Visualization of irrelevant data detection within the CIFAR100 dataset. The **left** column is one example of the target class and the images on the **right** columns are selected irrelevant data in the corresponding classes detected by LAVA.

## B.2  ABLATION STUDY

We perform an ablation study on validation size and on the hyperparameters in our method, where we provide insights on the impact of setting changes. We use the mislabeled detection use case and the CIFAR-10 dataset as an example setting for the ablation study.

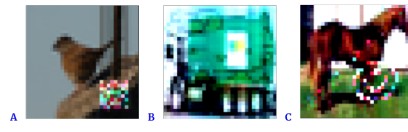

Figure 8: Visualization of each backdoor attack: A) Trojan-SQ attack. B) Blend attack. C) Trojan-WM attack.

### B.2.1  VALIDATION SIZE

For all the experiments in the main text, we use the validation set of size 10,000. Naturally, we want to examine the effect of the size of the validation set on the detection rate of mislabeled data. In Figure 9 (c), we illustrate the performance on the detection rate with smaller validation data sizes: 200, 500, 2,000, and 5,000. We observe that even reducing the validation set by half to 5,000 can largely maintain the detection rate performance. Small validation sets (200, 500, 2,000) degrade the detection rate by more than 50%. Despite the performance degradation, our detection performance with these small validation sizes is in fact comparable with the baselines in Figure 3 IV.(a) that

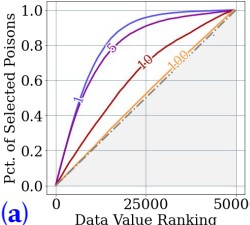 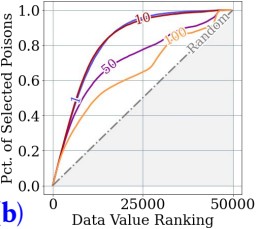 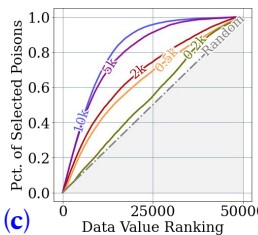

(a) (b) (c)

Figure 9: (a) Comparison between different feature weights on the performance of mislabeled data in CIFAR-10. (b) Comparison between different label weights on the performance of mislabeled data in CIFAR-10. (c) Comparison between different validation sizes on inspecting 50k samples from CIFAR-10 with $25\%$ mislabeled data.

leverage the full validation size of $10,000$. Additionally, when restricting LAVA and other baselines to validation set of 500 samples, our method is better than the best baseline for detecting mislabeled data in the 50k CIFAR-10 samples with $25\%$ being mislabeled as shown in Figure 11.

### B.2.2 FEATURE WEIGHT

Recall the class-wise Wasserstein distance is defined with respect to the following distance metric: $\mathcal{C}((x_t, y_t), (x_v, y_v)) = \mathrm{d}(x_t, x_v) + cW_{\mathrm{d}}(\mu_t(\cdot|y_t), \mu_v(\cdot|y_v))$. Actually, one can change the relative weight between feature distance $\mathrm{d}(x_t, x_v)$ and the label distance $W_{\mathrm{d}}(\mu_t(\cdot|y_t), \mu_v(\cdot|y_v))$. Here, we show the effect of upweighting the feature distance, while keeping the label weight at 1 and the results are illustrated in Figure 9 (a). As we are moving away from uniform weight, the performance on detection rate is decreasing with larger feature weights. With feature weight of 100, our method performs similarly as the random detector. Indeed, as we increase weight on the features, the weight on the label distance is decreased. As the weight reaches 100, our method performs similarly as the feature embedder without knowing label information and hence, the mislabeled detection performance is comparable to the random baseline.

### B.2.3 LABEL WEIGHT

Next, we shift focus to label weight. We examine the effect of upweighting the label distance, while keeping the feature weight at 1. In Figure 9 (b), as the label weight increases, the detection rate performance deteriorates. When we increase the label distance, the feature information becomes neglected, which is not as effective as the balanced weights between feature and label distances.

### B.2.4 FEATURE EMBEDDER

We use feature embedder to extract features for the feature distance part in our method. We train the feature embedder on the accessible validation set until the convergence on the train accuracy. Different architectures of the embedder might be sensitive to different aspects of the input and thus result in different feature output. Nevertheless, as we observe in Figure 10, the detection performance associated with different model architectures of feature embedder is similar. Hence, in practice, one can flexibly choose the feature embedder to be used in tandem with our method as long as it has large enough capacity. Furthermore, we note that that these feature embedders

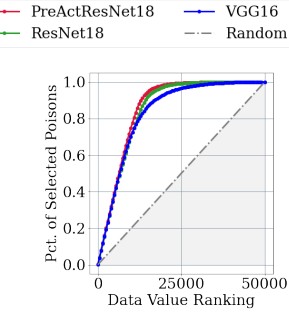

Figure 10: Comparison between different feature embedder architectures on inspecting 50k samples from CIFAR-10 with $25\%$ mislabeled data.

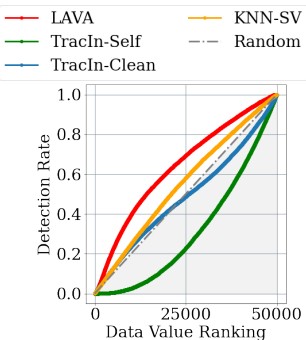

Figure 11: Detection rate by various methods on mislabeled CIFAR-10 using validation size of 500.

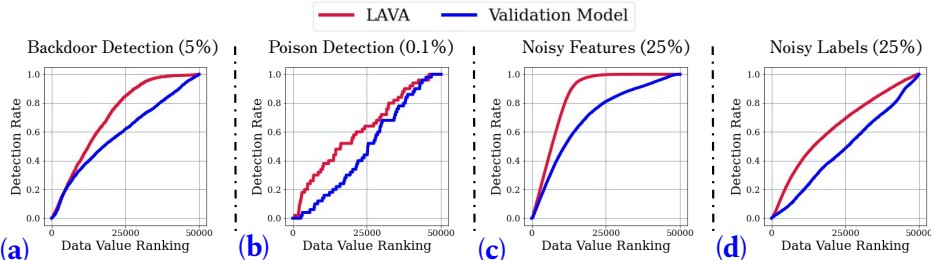

Figure 14: Detection performance comparison between LAVA and the model trained on validation data of size 500 on various use cases in CIFAR-10.

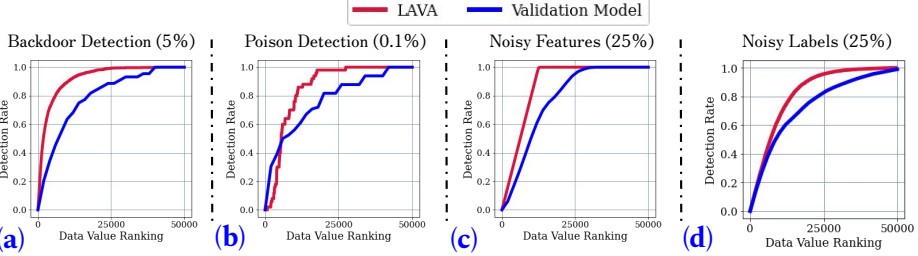

Figure 15: Detection performance comparison between LAVA and the model trained on validation data of size $10,000$ on various use cases in CIFAR-10.

have not learned the clean distribution from the validation data, e.g. in CIFAR-10 the model trained on $10K$ validation data achieves only around $65\%$ accuracy on $50K$ clean datapoints and the model trained on 500 validation data achieves around $25\%$ accuraracy. We additionally show in Figure 14,15 that our method significantly outperforms the PreActResNet18 model trained directly on validation data of size 500 and $10K$ in detecting bad data, which clearly distinguishes LAVA from simple feature embedders.

### B.3 Balancing Unbalanced Dataset

Although machine leaning practitioners might be using clean data for training a model, the dataset can be often unbalanced which can lead to model performance degradation (Thai-Nghe et al., 2009). To recover higher model accuracy, we can rebalance unbalanced datasets by removing points that cause such disproportion. We showcase how LAVA effectively rebalance the dataset by removing points with poor values and keeping points with best values. We consider a CIFAR-10 dataset with a class *Frog* being unbalanced and containing $5,000$ samples while other classes have only half as much (i.e. $2,500$ samples). In Figure 12, we demonstrate the effectiveness of LAVA valuation which not only reduces the dataset by removing poor value points but also improves the model accuracy. While at the same time other valuation methods were not able to steadily increase the model accuracy and quickly downgraded the model performance, which in turn shows an even stronger effectiveness of our method.

### B.4 Reducing Training Set Size

With the growing size of the training dataset, the computation cost and memory overhead naturally increase which might deem impos-

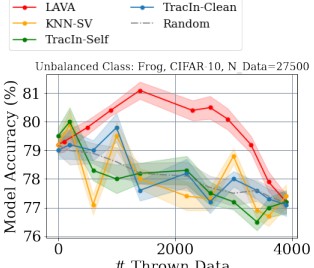

Figure 12: Comparison of various methods on rebalancing an unbalanced dataset on CIFAR-10 with the class *Frog* being unbalanced.

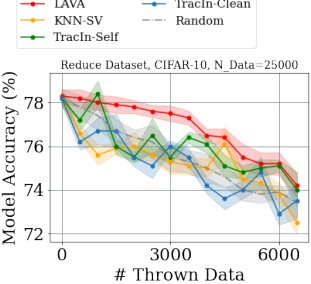

Figure 13: Comparison of various methods on reducing a dataset size based on valuation of datapoints on CIFAR-10.

sible for some practitioners with limited resources to train a model. Therefore, the ability to reduce the training dataset size (Sener & Savarese, 2018) will free up some computation burden and thus allow ones with limited resources to fully appreciate the model training process. Motivated by the given challenge, we want to leverage our data valuation method to significantly decrease the training dataset size while maintaining the model performance. Similarly as in the previous section, the idea is to keep a subset of datapoints with best values and remove poor valued ones. To demonstrate the effectiveness of our LAVA's valuation, we perform such a task on a clean CIFAR-10 dataset with $2,500$ samples from each class and compare with other data valuation methods. As presented in Figure 13, it is demonstrated that the performance is well maintained even with smaller subsets of the original dataset. Remarkably, even reducing a clean training set (25,000 samples) by $15\%$ based on our method's valuation, the performance can still stay relatively high while outperforming other valuation baselines.

### B.5    DATA SUMMARIZATION

With growing dataset sizes, grows the space needed to store data. Thus, the buyer often would like to decrease the dataset to minimize resources but to retain the performance. Unlike reducing training set size as provided in Section B.4, in this experiment, we will select a smaller, representative subset of the whole dataset that can maintain good performance. To measure the performance of each subset, we measure the validation performance of the model trained on that subset subtracted by the validation performance of the model trained on a random subset of the same size, the experiment which is performed in Kwon & Zou (2021). In Figure 16 we can observe that our method can select a small subset that performs better than the subsets chosen by the baseline methods most of the time.

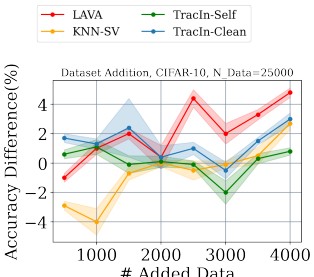

Figure 16: Comparison of various methods on data summarization based on valuation of datapoints on CIFAR-10.

### B.6    SCALABILITY EXPERIMENT

In the main paper, we have demonstrated time complexity comparison between LAVA and other valuation methods. We have reported runtime comparisons only for 2,000 test samples as this is the scale existing methods can solve in a not excessively long time (within a day). It showcases the advantageous computing efficiency that the proposed approach enjoys over other methods. We further want to emphasize the computational efficiency of LAVA and demonstrate computation efficiency on a larger scale dataset (100,000 samples) with higher dimensions, ImageNet-100. Additionally, we evaluate

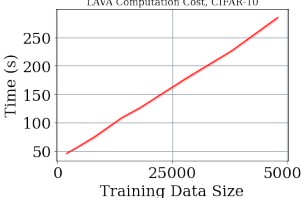

Figure 17: Near-linear time complexity of LAVA shown on CIFAR-10.

other baselines which are able to finish within a day of computation to highlight the advantage of our method as presented in Table 1. Moreover, we highlight the near-linear time complexity of LAVA on CIFAR-10, which shows practical computation efficiency of our method as shown in Figure 17.

### B.7    GENERALIZATION TO OTHER TYPES OF BACKDOOR ATTACKS

As we have provided the results of the Trojan square attack (Trojan-SQ) (Liu et al., 2017) in Section 4, we now apply LAVA to other backdoor attacks, which are Hello Kitty blending attack (Blend) (Chen et al., 2017) and Trojan watermark attack (Trojan-WM) (Liu et al., 2017), and evaluate the efficacy of our method in detecting different types of backdoor attacks. We simulate these attacks by selecting the target class *Airplane* and poisoning $2,500$ $(5\%)$ samples of the CIFAR-10 dataset of size $50,000$. The backdoor trigger adopted in each attack is portrayed in Figure 8. In Figure 18, we observe that our method can achieve superior detection performance on all the attacks considered. The reason is that despite the difference in trigger pattern, all of these attacks modify both the label and the feature of a poisoned image and thus result in the deviation of our distributional distance that is defined over the product space of feature and label.

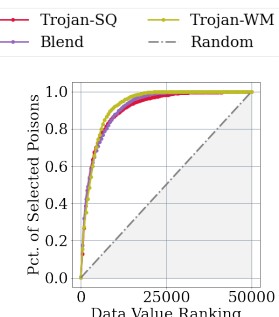

Figure 18: Detection rate of various backdoor attacks by LAVA.

## B.8 IMPLICATIONS OF THE PROPOSED DATA VALUATION METHOD TO REAL-WORLD DATA MARKETPLACES

| Method | Time |
|---|---|
| LAVA | 1 hr 54 min |
| KNN-SV | 4 hr 21 min |
| TracIn-Clean | 7 hr 50 min |
| TracIn-Self | 7 hr 51 min |

Table 1: Comparison of runtime between various methods needed to valuate ImageNet-100.

One concern in the real-world data marketplace is that data is freely replicable. However, replicates of data introduce no new information and therefore the prior work has argued that a data utility function should be robust to direct data copying (Xu et al., 2021). One advantage of using the class-wise Wasserstein distance to measure data utility is that it is *robust to duplication*. Our method by its natural distributional formulation will ignore duplicates sets. As shown in Table 3, although we have repeated the set even five times more than the original source set, the distance remains the same. Additionally, with small noise changes in the features, the distance metric is barely affected. Another concern in the real-world marketplace is that one might find a single data that has highest contribution and duplicate it to maximize the profit. However, again due to the nature of our distributional formulation, duplicating a single point multiple times would increase the distance between the training and the validation set due to the imbalance in training distribution caused by copying that point.

## B.9 DETAILED EXPERIMENTAL SETTINGS

**Datasets and Models.** Table 2 summarizes the details of the dataset, the models, as well as their licenses adopted in our experiments.

**Hardware.** A server with an NVIDIA Tesla P100-PCIE-16GB graphic card is used as the hardware platform in this work.

**Software.**

For our implementation, we use `PyTorch` for the main framework (Paszke et al., 2019), assisted by three main libraries, which are `otdd` (optimal transport calculation setup with datasets) (Alvarez-Melis & Fusi, 2020), `geomloss` (actual optimal transport calculation) (Feydy et al., 2019), and `numpy` (tool for array routines) (Harris et al., 2020).

| Dataset Size | OT Dist | |
|---|---|---|
| | Direct | Near |
| 5000 | 195.64 | +0.98 |
| $2 \times 5000$ | +0.00 | +0.98 |
| $3 \times 5000$ | +0.00 | +0.98 |
| $4 \times 5000$ | +0.00 | +0.98 |
| $4.5 \times 5000$ | +0.07 | +0.98 |
| $5 \times 5000$ | +0.00 | +0.98 |

Table 3: Class-wise Wasserstein distance behavior under dataset direct duplication and its near duplicates.

Table 2: Summary of datasets and models and their licenses used for experimental evaluation. (Note: "Detail" column implicitly refers to training data, unless explicitly noted.)

| Experiment Type | Dataset | License | Model | License | Train/ Valid Size | Detail |
|---|---|---|---|---|---|---|
| Wasserstein vs Model Performance (Fig 1) | STL10 | Not Applicable | GoogLeNet | MIT | 5K/8K | 0%, 2%, 5%, 10%, 15% Mislabeled |
| | CIFAR10 | MIT | ResNet18 | MIT | 50K/10K | 0%, 2%, 5%, 10%, 15% Mislabeled |
| | CIFAR100 | MIT | ResNet50 | MIT | 50K/10K | 0%, 2%, 5%, 10%, 15% Mislabeled |
| Backdoor: Trojan Sq (Fig 3) | CIFAR10 | MIT | ResNet18 | MIT | 50K/10K | 5% Poisoned |
| Poisoning: Collision (Fig 3) | CIFAR10 | MIT | ResNet18 | MIT | 50K/10K | 0.1% Poisoned |
| Noisy Feature (Fig 3) | CIFAR10 | MIT | ResNet18 | MIT | 50K/10K | 25% Noisy Data |
| Noisy Labels (Fig 3) | CIFAR10 | MIT | ResNet18 | MIT | 50K/10K | 25% Noisy Data |
| Irrelevant Data (Fig 4) | CIFAR10 | MIT | PreAct-ResNet18 | MIT | 50K/10K | 1 Target Class Spread Into 9 Other Classes |
| Runtime VS Perf (Fig 5) | CIFAR10 | MIT | ResNet18 | MIT | 2K/1K | 10% Backdoors |
| Backdoor: Blend (Fig 6) | GTSRB | CC0 1.0 | PreAct-ResNet18 | MIT | 35888/ 12360 | 5% Poisoned |
| Noisy Feature (Fig 6) | MNIST | CCA-SA 3.0 | PreAct-ResNet18 | MIT | 60K/10K | 25% Noisy Data |
| Time Complexity (Table 1) | ImageNet-100 | Not Applicable | ResNet50 | MIT | 100K/10K | 25% Mislabeled 10 Classes |
| Irrelevant Data (Fig 7) | CIFAR100 | MIT | PreAct-ResNet34 | MIT | 50K/10K | 1 Target Class Spread Into 99 Other Classes |
| Noisy Label (Fig 10) | CIFAR10 | MIT | PreAct-ResNet18 | MIT | 50K/10K | 25% Noisy Label |
| | | | VGG16 | CC BY 4.0 | | |
| | | | ResNet18 | MIT | | |
| Noisy Label (Fig 9) | CIFAR10 | MIT | PreAct-ResNet18 | MIT | 50K/10K | Different Feature Weights 1,5,10,100 Different Label Weights 1,10,50,100 |
| | | | | | 50K/10K 50K/5K 50K/2K 50K/0.5K 50K/0.2K | Different Validation Sizes |
| Backdoor: Blend, Trojan, SQ-WM (Fig 18) | CIFAR10 | MIT | Not Applicable | Not Applicable | 50K/10K | Visualization of Attacks Detection Rate of Lava |
| Data Duplication (Table 3) | Not Applicable | Not Applicable | Not Applicable | Not Applicable | 5K/5K | Duplication of Training Set |
| Dataset Reduction (Fig 13) | CIFAR10 | MIT | PreAct-ResNet18 | MIT | 25K/10K | 2.5K Samples From Each Class |
| Data Summarization (Fig 16) | CIFAR10 | MIT | PreAct-ResNet18 | MIT | 25K/10K | 2.5K Samples From Each Class |
| Unbalanced Dataset (Fig 12) | CIFAR10 | MIT | PreAct-ResNet18 | MIT | 27.5K/10K | 5K Samples From Class Frog 2.5K Samples From Other Classes |
| Time Complexity (Fig 17) | CIFAR10 | MIT | PreAct-ResNet18 | MIT | 50K/10K | 5K From Each Class |

