# OpenReview forum: "LAVA: Data Valuation without Pre-Specified Learning Algorithms"
_ICLR.cc/2023/Conference — ICLR 2023 notable top 25%_

### Official Review · Reviewer_5BsF · 2022-10-14

**Confidence:** 2
**Correctness:** 2
**Technical Novelty And Significance:** 3
**Empirical Novelty And Significance:** 2
**Recommendation:** 8

**Clarity, Quality, Novelty And Reproducibility:**

Significance: The problem of data valuation is an important and contemporary problem in machine learning. The paper proposes a method to decouple the data valuation process from the learning algorithm. This is a meaningful contribution and is suitable to be presented at ICLR 2023.

Soundness: The paper proposes new solutions to the data valuation problem. However, I feel that these solutions are not satisfactorily justified. The main theoretical result (Theorem 1) seems to be very vacuous and follows almost immediately from the Lipchitzness assumption. Please refer to the section below for more detailed comments.

Novelty: The paper incorporates concepts from optimal transport to the problem of data valuation, offering an unconventional alternative to the common Shapley value (or more broadly cooperative game theory) approaches to data valuation.

Presentation: The paper is generally well-written and easy to follow. However, the authors occasionally make some arguments that are hard to follow or not well justified by the evidence. Please refer to the section below for more detailed comments.

**Strength And Weaknesses:**

Strengths:

(+) The paper is sufficiently novel and the approach is interesting.

Weaknesses:

(-) The proposed approach lacks justification.

(-) The experiment setups are unclear.

(-) The presentation can be improved.

Detailed comments:

- (main concern - soundness) Theorem 1 resembles a statistical learning generalization bound, where we have an out-of-sample error (in this case, the validation set’s error) upper bounded by the training set’s error and some surplus term (the OT distance). By arguing that the training set’s error is close to 0 for sufficiently complex models, the authors justify the use of OT distance as a surrogate for the training set’s error by this theorem. However, one can apply the same logic to other generalization bounds (e.g., one based on VC dimension) to justify the use of VC dimension as a surrogate for the validation error. I find that this theorem does not satisfactorily justify the use of OT distance as a surrogate for the validation error.

- (main concern - soundness) Theorem 1 is vacuous and follows almost immediately from the assumptions that $\mathcal{L}$ is $k$-Lipschitz and the norm of $f$ is bounded by V. Could the authors elaborate on the challenges/difficulty in the proof of Theorem 1.

- The authors did not explain the experiment setups (not even in the appendix). What is the validation dataset here? And in the mislabeled or noisy feature experiments, are the validation data corrupted too? It is unclear how the proposed framework is applied in these experiments.

- (presentation) In the paragraph “Radius for accurate predictions…”, can the authors be more specific about the results in Bertsimas & Tsitsiklis (1997). I find this paragraph a bit hand-wavy.

- (presentation) I do not follow the argument in this sentence (at the beginning of Section 3.1): “OT distance is known to be insensitive to small differences while also being not robust to large deviations”. I think the phrase “not robust” is not appropriate here. Also, I’m not sure why that is a desirable property in data valuation.

- (presentation) Please use more equation numbering: Throughout the paper, the authors refer to various quantities/definitions via section number. This is very challenging for readers to follow. Could you please put important quantities/definitions in equation environments with numbers and refer to them?

- (minor aesthetic comment - presentation) The results in Figures 2 and 3 are blurry. Although I can still understand the results, it would be nicer if the authors can include clearer versions of these plots.

**Summary Of The Paper:**

The paper proposes a learning-agnostic data valuation framework based on optimal transport (OT). The authors suggest using a Class-wise Wasserstein distance between two datasets as a surrogate for validation performance, thus alleviating the need to pre-specifying the learning algorithm in advance. The authors then propose a new data valuation metric based on the gradient of the OT distance with respect to the perturbation on the probability mass associated with each data point. Even though this data valuation metric is difficult to compute, the paper points out that the difference can be calculated efficiently, allowing for an efficient ranking of the data points. The paper then presents empirical results to demonstrate the efficiency of the proposed methods on several different problems.

**Summary Of The Review:**

The paper offers a fresh perspective on the problem of data valuation using ideas from optimal transport. However, the proposed framework is not sufficiently justified, both theoretically and empirically. The presentation of the paper can be improved.

======= Post Rebuttal =======

The authors address some of my concerns and I have increased the score accordingly.

---

> ### Author Response · Authors · 2022-11-14
> **(minor 5) Figures 2 and 3 are blurry**
>
> > *"(minor aesthetic comment - presentation) The results in Figures 2 and 3 are blurry. Although I can still understand the results, it would be nicer if the authors can include clearer versions of these plots."*
>
> **Re:** We are sorry for the figures’ quality. We have included figures with higher resolution.

---

> ### Author Response · Authors · 2022-11-14
> **(minor 4) Equation indexing**
>
> > *"(presentation) Please use more equation numbering: Throughout the paper, the authors refer to various quantities/definitions via section number. This is very challenging for readers to follow. Could you please put important quantities/definitions in equation environments with numbers and refer to them?"*
> >
> **Re:** We agree with the suggestion and will update the numbering rule to ease the experience for the readers.

---

> ### Author Response · Authors · 2022-11-14
> **(minor 3) "OT distance is known to be insensitive to small differences while also being not robust to large deviations” as a desirable property for data valuation**
>
> > *"(presentation) I do not follow the argument in this sentence (at the beginning of Section 3.1): “OT distance is known to be insensitive to small differences while also being not robust to large deviations”. I think the phrase “not robust” is not appropriate here. Also, I’m not sure why that is a desirable property in data valuation."*
>
> **Re:** We note the OT distance (2-Wasserstein distance) being "insensitive to small differences" in the sense that **the distance changes little w.r.t. small wiggles in the distribution**. In the scenario of data valuation, it means we can expect **the OT distance not to focus on the small variations between "normal" data points (in-distribution data).** We also have some results in Appendix B.9, Page 21 that may serve as an empirical verification of this. On the other hand, we note the OT distance is "not a robust metric" as **it is not robust to large noises/deviations–if there's a probability mass having a large distance in the distribution, the OT distance can be arbitrarily large depending on this deviation–despite that the probability mass may be small**. More precisely, the term "not robust" refers to the fact that the OT distance tends to infinity if there is noise/deviation that goes to infinity in the distribution distance. In the case of data valuation, **we expect this allows the OT distance to be more sensitive to abnormal data points that are associated with larger deviations in the distribution (out-of-distribution data).**
>
> The detailed tutorial may be helpful here (incl. discussions on "being insensitive to small wiggles in the distribution" in Sec. 1.6 and "not being robust to large deviations" in Sec. 7), and it points to more rigorous references about this property.
> > “Optimal Transport and Wasserstein Distance”, https://www.stat.cmu.edu/~larry/=sml/Opt.pdf
>
> **In general, we found these properties to be well-aligned with what we are looking for in the scenario of data valuation that has motivated the development of this work.** We are appreciative of the helpful discussion and will include the explanations in the updated version.

---

> ### Author Response · Authors · 2022-11-14
> **(minor 2) Details about the result of “Radius for accurate predictions…” provided in Bertsimas & Tsitsiklis (1997)**
>
> > *"(presentation) In the paragraph “Radius for accurate predictions…”, can the authors be more specific about the results in Bertsimas & Tsitsiklis (1997). I find this paragraph a bit hand-wavy."*
>
> ***TL;DR:*** *The methodology we adopted to calculate the gradients of the OT distance w.r.t. the change of probability mass for the underlying distributions follows the standard approach for sensitivity analysis w.r.t. the parameter change in linear programming. Here, we provide detailed references for how the work is constructed, and also a conceptual introduction and discussions for the underlying mathematical derivations and the development of the methods. We will compile it into a new section in the Appendix for readers' reference.*
>
> **Re:** **The proposed method that uses dual solutions to construct the predictor for the change in the OT distance and the development of the calibrated gradient (Eq. 1) is based on the idea of using optimal dual variables as marginal costs**, where a tutorial introduction with discussion can be found in *Chapter 4.4, Optimal dual variables as marginal costs, Page 155-156, Bertsimas & Tsitsiklis (1997)*. **For deriving the range where this prediction holds precisely and discussions on what happens beyond the range, we follow the principled approach for local sensitivity analysis w.r.t. parameter changes**, for which a tutorial introduction with discussion can be found in *“Changes in the requirement vector b”, Chapter 5.1, Local sensitivity analysis, Page 207-208, Bertsimas & Tsitsiklis (1997)*.
>
> **The conceptual idea is that when we perturb the probability mass, which is the right-hand side of the constraints in the primal formulation, we examine whether the dual solutions remain optimal–if it is still the optimal solution, then the prediction we obtain is precise; otherwise, the prediction will be approximate.** In the dual formulation, the probability mass vector becomes the cost vector–no longer in the constraints. Thus, the previously obtained dual optimal solution will always be feasible when perturbing this parameter as the dual constraints are not changed *(ref: Chapter 4.2, The dual problem, Page 142-146, Bertsimas & Tsitsiklis (1997))*. From the duality theorem, we know that any dual feasible solution is primal optimal if it is primal feasible (which is not necessarily the case). Thus, if the solution is also primal feasible, then it is the *optimal* solution. *(ref: Theorem 4.3 (Weak duality), Page 146, Bertsimas & Tsitsiklis (1997); Corollary 4.2, Page 148, Bertsimas & Tsitsiklis (1997))*.
>
> **Back to our case, when we perturb the parameter (probability mass), the dual solution we obtain from the OT solver always remains dual feasible. We want to know if it also remains primal feasible.** If so, we know the dual solution is still optimal despite the perturbation, and thus the prediction we constructed using the dual solution is precise. The range (the radius for precise predictions) we obtained is calculated as follows. **Given the basic variables of the primal formulation corresponding to the dual solution, when perturbing the right-hand side of the primal constraints (probability mass), we want to know how far we can go under these basic variables before any constraint is violated.** This is the range where the prediction is precise. When we go beyond the range, there will be primal constraints being violated, which means the primal problem is no longer feasible under these basic variables and the dual solution is no longer optimal (despite still being dual feasible). This is the case where our prediction is reduced to an approximation. Principally, under such perturbation, the previously-optimal dual solution is still near the new optimal solution even if it is no longer optimal–most primal constraints are still satisfied and it is still *mostly* primal feasible. Since the dual solution still encodes partial information, we can obtain nice approximations.
>
> We thank the reviewer for pointing out this potentially unclear discussion in the paper. We hope this answer helps explain the work. Rigorous mathematical derivations and detailed procedures for calculations can also be found in the aforementioned references. We think these discussions and explanations can bring additional insight to the readers and we will include them in the Appendix.
>
> > Bertsimas, D., & Tsitsiklis, J. N. (1997). Introduction to linear optimization (Vol. 6, pp. 479-530). Belmont, MA: Athena Scientific.

---

> > ### Comment · Reviewer_5BsF · 2022-11-19
> > **Response**
> >
> > Thank you, this helps a lot! I understand now how we can obtain the calibrated gradient directly from the OT solver.

---

> ### Author Response · Authors · 2022-11-14
> **(minor 1) Experimental setup**
>
> > *"The authors did not explain the experiment setups (not even in the appendix). What is the validation dataset here? And in the mislabeled or noisy feature experiments, are the validation data corrupted too? It is unclear how the proposed framework is applied in these experiments."*
>
> **Re:** We apologize for not providing the full experimental setup. **Validation dataset is the model/target dataset that the buyer would desire the training data to align with and the training/given datasets are the data to be evaluated. For this reason, we assume that the validation dataset contains only clean data, which is a common assumption in data valuation literature.** For all experiments, we use clean validation datasets. In detecting bad data experiments, we prepare a corrupted training dataset and a clean validation dataset. The goal is to identify the bad data in the training dataset. First, LAVA measures the class-wise OT distance between two datasets. Second, it calculates sensitivity analysis to get the values of the training data points. Then, we rank the points based on their values and verify the effectiveness through detection and model performance. We will add a detailed subsection for experimental setup in the updated version. Additionally, we already have a table with all datasets and model settings presented in Table 2.

---

> ### Author Response · Authors · 2022-11-14
> **(major 2) Importance, challenges and difficulty of Theorem 1**
>
> > "(main concern - soundness) Theorem 1 is vacuous and follows almost immediately from the assumptions $\mathcal{L}$ that is $k$-Lipschitz and the norm of $f$ is bounded by V. Could the authors elaborate on the challenges/difficulty in the proof of Theorem 1.""
>
>
> ***TL;DR:*** *Due to the (long) definitions of notations, statement of assumptions, derivations for certain conditions, and the arduous procedure to develop the theorem, to retain a relatively easy reading experience and have a smooth presentation of the conceptual flow of the work, **we had to postpone the complete presentation of the theorem and its full proof to the Appendix (A.1-A.3).***
>
> *Theorem 1 derives an upper bound depending on the notion of the hierarchically-defined Wasserstein distance *grounded on the hybrid cost* in a form similar to the celebrated Kantorovich-Rubinstein duality for conventional Wasserstein distance. **The novelty of the proof lies in a series of challenging mathematical constructions to make the hierarchically-defined hybrid Wasserstein distance ultimately appear on the upper bound, thus providing this notion a formal justification.** The construction of the proof procedure is the original contribution of this work, not trivially adapted from existing or similar results.*
>
> **Re:** We thank the reviewer for sparing attention to the construction of the main theorem in this work, and we would like to take this opportunity to share the challenging but rewarding process of discovering this interesting result.
>
> **The celebrated Kantorovich-Rubinstein Duality (KR-duality) provided the essential theoretical foundation for the Wasserstein distance grounded on the Euclidean distance (2-Wasserstein distance)** capping the difference between training and validation loss in expectation under Lipschitz conditions for the prediction model. As mentioned in the response above, rather than the conventional 2-Wasserstein distance, **in this work, we use the novel notions of class-wise optimal transport and hierarchically-defined hybrid Wasserstein distance**, which consider not only data values but also labels. The Wasserstein distance with this particular hybrid cost has shown great empirical performance in detecting abnormal features and labels in our experiments. However, all of these observations remain at an empirical level. **This paper provides the first theoretical justification for this hierarchically-defined hybrid Wasserstein distance metric in a form similar to the bounds derived from KR-duality for conventional 2-Wasserstein distance, filling the blank in current research and bridging the gap between theoretical understanding and practical applications.**
>
> It is straightforward to see that the formulations for the class-wise optimal transport and the corresponding hierarchically-defined Wasserstein distance grounded on the special hybrid cost (Eq. 5 in Appendix A.2) are notably distinct from those of the conventional optimal transport and the 2-Wasserstein distance. The adaptation from the results provided in KR-duality to what is desired for the novel notion is nontrivial. **We note the exploration process to find the appropriate assumptions (e.g. probabilities Lipschitz conditions, introduced in Appendix A.2) and the combination of sophisticated techniques are considerably involved and challenging. The construction of the proof procedure is the original contribution of this work, not trivially adapted from existing or similar results. The arduous procedure for deriving the conditions and constructing the terms can be seen in our full proof in Appendix A.3.**
>
> Ultimately, we feel fortunate that we made it through and the efforts led to the discovery of this much-needed result. We note this contribution to be novel, unique, and important. Please let us know if this reply has addressed Your concerns and we would like to be glad to share more about the development of this work.
>
> > J Thickstun, "Kantorovich-Rubinstein Duality",  https://courses.cs.washington.edu/courses/cse599i/20au/resources/L12_duality.pdf

---

> > ### Comment · Reviewer_5BsF · 2022-11-19
> > **Response**
> >
> > > The construction of the proof procedure is the original contribution of this work, not trivially adapted from existing or similar results.
> >
> > Since the authors mention that the proof is completely original and not adapted from existing works, I agree that the discovery process to establish the appropriate assumptions in itself is challenging.
> >
> > Now that I take a closer look at the proof, there may be an issue. We impose probabilistic Lipschitz assumptions on $f_t$ and $f_v$, so there's an error probability of $\delta$. However, when this property is used, it is used as if the Lipschitz holds almost surely.
> >
> > To fix this in the proof, I would expect a union-bound somewhere and expect the final result to be probabilistic with an error rate scaled with the number of data points.
> >
> > I recognize that this is the end of the rebuttal period and apologize for the late response. I raise this as a potential issue and do not expect the authors to rectify it. For the purpose of this review, I'll ignore this issue.

---

> > > ### Author Response · Authors · 2022-12-13
> > > **probabilistic notion in the final result**
> > >
> > > **Why the final results are not probabilistic given that probabilistic Lipschitz conditions are assumed?**
> > >
> > > > "...We impose probabilistic Lipschitz assumptions... so there's an error probability... However, when this property is used, it is used as if the Lipschitz holds almost surely...would expect a union-bound somewhere and expect the final result to be probabilistic with an error rate scaled with the number of data points..."
> > >
> > > **Re:** The authors are appreciative of the reviewer for taking the effort to help with the development of the work and raise the interesting question. We think this is an excellent question as it touches on some conceptual ideas of the proof as well as the relevant techniques.
> > >
> > > On a high level, the probabilistic cross-Lipschitz condition states, with a high probability, outputs of two functions are close if their respective inputs are close. In this case, the probabilistic cross-Lipschitz condition assumes that with a high probability, the label of a datapoint in the training dataset is close to the label of a datapoint in the validation dataset if these two datapoints are close in the input space, (assuming the labels of datapoints take continuous value.) Intuitively, the condition bounds the probability of finding pairs of training and validation data with close features(–i.e., in a local region) but are labeled differently.
> > >
> > > When dealing with this condition in the proof, we first separate the input space into regions where the (deterministic) Lipschitz condition does hold and where it does not. For the region where the (deterministic) Lipschitz condition holds, we continue the derivation with the (deterministic) Lipschitz condition; for the region where (deterministic) Lipschitz condition does not hold, we directly assume the labels of the training data and validation data are *different* regardless of their features(–i.e., are close or not). Finally, we combine these two parts into the expectation and attain an *upper bound* of the generalization error, where the last term (constant term) of the final result in the theorem represents the part where the condition does not hold.
> > >
> > > In short, the probabilistic notion translates into the upper bound in the final result. Such treatment can be found in research on slightly different topics, such as bounding the target error in domain adaptation
> > >
> > > > Courty, N., Flamary, R., Habrard, A., & Rakotomamonjy, A. (2017). Joint distribution optimal transportation for domain adaptation. Advances in Neural Information Processing Systems, 30.
> > > https://proceedings.neurips.cc/paper/2017/file/0070d23b06b1486a538c0eaa45dd167a-Paper.pdf
> > >
> > > We thank the reviewer for pointing out this interesting part and we enjoy the insightful discussion.

---

> ### Author Response · Authors · 2022-11-14
> **(major 1) Soundness 3/3**
>
> (Cont'd)
>
> **iii. What is novel in this work/why we believe this is an important and unique contribution to the existing research and tools?**
>
> ***TL;DR:** There is an absence of theoretical justifications for the novel class-wise OT and hierarchically-defined hybrid Wasserstein distance despite it being a promising tool. This work is the first to fill the blank, providing its much-needed theoretical foundations and also extending it to attractive applications.*
>
> **Re:** **The development of our methods is not based on the conventional optimal transport/Wasserstein distance between empirical distributions of data values, rather, we use the novel notions of class-wise optimal transport and hierarchically-defined hybrid Wasserstein distance, which consider not only data values but also the labels.** The Wasserstein distance with this particular hybrid cost has shown great empirical performance in detecting abnormal features and labels in our experiments. In addition, this Wasserstein distance has also been shown in past work [2] to characterize the domain and task difference. However, all of these observations remain at an empirical level. **This paper provides the first theoretical justification for the Wasserstein distance with the hybrid cost by showing that it characterizes the upper bound of the validation performance.** Our proof might be of independent interest to a broader community involving meta-learning and domain adaptation as it can be simply adapted to provide justification for why this Wasserstein distance is a good proxy for predicting transfer learning and meta-learning performances. The novelty of the proof lies in a series of mathematical constructions to make the Wasserstein distance grounded on the hybrid cost appear on the upper bound, thus providing this Wasserstein distance a formal justification. *(Detailed breakdown discussions on the technical challenges of developing the theorem and our contributions are provided in the reply to Major Concern 2 below.)*
>
> We hope that this long answer can address your concern. Please let us know if you have further comments.
>
> > [1] Kullback, Solomon, and Richard A. Leibler. "On information and sufficiency." The annals of mathematical statistics 22.1 (1951): 79-86. \
> > [2] Rudin, Leonid I., Stanley Osher, and Emad Fatemi. "Nonlinear total variation based noise removal algorithms." Physica D: nonlinear phenomena. 1-4 (1992): 259-268. \
> > [3] Gabor J Szekely, Maria L Rizzo, et al. Hierarchical clustering via joint between-within distances: Extending ward’s minimum variance method. Journal of classification, 22(2):151–184, 2005 \
> > [4] David Alvarez-Melis and Nicolo Fusi. Geometric dataset distances via optimal transport. Advances in Neural Information Processing Systems, 33:21428–21439, 2020. \
> > [5] Jean Feydy, Thibault Séjourné, François-Xavier Vialard, Shun-ichi Amari, Alain Trouvé, and Gabriel Peyré. Interpolating between optimal transport and mmd using sinkhorn divergences. In The 22nd International Conference on Artificial Intelligence and Statistics, pages 2681–2690. PMLR, 2019. \
> > [6] Aude Genevay, Gabriel Peyré, and Marco Cuturi. Learning generative models with sinkhorn divergences. In International Conference on Artificial Intelligence and Statistics, pages 1608–1617. PMLR, 2018. \
> > [7] Villani, Cédric (2009). Optimal Transport, Old and New. Grundlehren der mathematischen Wissenschaften. Vol. 338. Springer-Verlag Berlin Heidelberg. p. 10. doi:10.1007/978-3-540-71050-9. ISBN 978-3-540-71049-3

---

> ### Author Response · Authors · 2022-11-14
> **(major 1) Soundness 2/3**
>
> (Cont'd)
>
> **ii. Why do we use OT/Wasserstein distance instead of other measures?**
>
> ***TL;DR:** Most of the frameworks for learning bounds (e.g, VC, Rademacher) cannot give similar results as presented in this paper, which measures the effects of data rather than of the model.*
>
> **Re:** There are, indeed, other discrepancy measures, such as Kullback-Leibler Divergence [1], Total Variation norm [2] or Maximum Mean Discrepancies (MMD) [3], that may work similarly to the optimal transport/Wasserstein distance and be able to bound the validation errors. However, **the mathematically well-defined OT distance has advantageous analytical properties (is a valid metric; compatible with sparse-support distributions; stable with respect to deformations of the distributions’ supports), and it is also computationally tractable and commutable from finite samples [4,5,6].**
>
> Specifically, Wasserstein distance is mathematically well-defined and provides desired analytical properties in many aspects. For example, KL-divergence generally requires that the discrete probability distributions be defined on the same probability space to calculate the relative entropy. Yet, in this work, when measuring the distributional discrepancy between the datasets, we treat each dataset directly as the empirical distribution of its data points, rather than approximating it with additional assumptions (e.g. Gaussian). If the data points are images, then each non-identical image is located in a different place in the high-dimensional space. KL-divergence between such data points is inappropriately defined and may not be a proper metric, while the Wasserstein distance provides better support for conducting such analysis. Moreover, Wasserstein distance is also favored from practical considerations. It is generally considered insensitive to small variations in distributions while also not robust to large deviations [7], making it particularly suitable for the data-valuation scenario. More specifically, the OT distance may ignore normal variations within clean data while being sensitive to abnormal data with large distributional deviations–which was empirically verified in this work. *(Detailed explanations on this property are provided separately in the reply to Minor Comment 3 below.)*

---

> ### Author Response · Authors · 2022-11-14
> **(major 1) Soundness 1/3**
>
> **1. Theorem 1 resembles a statistical learning generalization bound.\
> 2. The same logic can be applied to other generalization bounds (e.g., one based on VC dimension) to justify the use of other measures as a surrogate for the validation error.\
> 3. The use of OT as a surrogate for the validation error is not sufficiently justified by this theorem.**
>
> > *"Theorem 1 resembles a statistical learning generalization bound, where we have an out-of-sample error (in this case, the validation set’s error) upper bounded by the training set’s error and some surplus term (the OT distance). By arguing that the training set’s error is close to 0 for sufficiently complex models, the authors justify the use of OT distance as a surrogate for the training set’s error by this theorem. However, one can apply the same logic to other generalization bounds (e.g., one based on VC dimension) to justify the use of VC dimension as a surrogate for the validation error. I find that this theorem does not satisfactorily justify the use of OT distance as a surrogate for the validation error."*
>
> ***TL;DR:***
> 1. *The central problem of the work (distribution discrepancy) is distinct from generalization bounds.*
> 2. *Most of the frameworks for learning bounds (e.g, VC, Rademacher) cannot give similar results as presented in this paper, which measures the effects of data rather than of  the model.*
> 3. *There is an absence of theoretical justifications for the novel class-wise OT and hierarchically-defined hybrid Wasserstein distance despite it being a promising tool. This work is the first to fill the blank, providing its much-needed theoretical foundations and also extending it to attractive applications.*
>
> **Re:** Thank You for the comment and for pointing out this potential problem of confusion.
>
> In this work, we rely on optimal transport and Wasserstein distance as the discrepancy measure to propose using the Wasserstein distance between the given data and the target data for data valuation. We justify the use with the theorem stating that the class-wise hybrid Wasserstein distance upper-bounds the prediction error on the validation data, and thus can be used as a predictor of the performance of models trained on the training data without actually training a model.
>
> Here, we answer the following three questions:
>
> **i.** *What is the central problem in this work/what do we use the discrepancy measures for?* \
> **ii.** *Why do we use OT/Wasserstein distance instead of other measures?* \
> **iii.** *What is novel in this work/why do we believe this is an important and unique contribution to the existing research and tools?*
>
> **i.** In this work, discrepancy measures between distributions are used to construct an indicator to predict the target validation performance of downstream models trained on the given training dataset without actually training the model. Specifically, **we treat the target (validation) and given (training) datasets as empirical distributions composed of their underlying data points. Then, we calculate the class-wise optimal transport distance between the distributions and use it as our predictor.**
>
> The usage of discrepancy measures here is different from the case in deriving generalization bounds. **Most of the frameworks used in deriving generalization bounds, such as the VC-dimension or Rademacher complexities, measure the richness of a class of real-valued functions (model classes). These measures are independent of data and thus fundamentally cannot be applied to the task of data valuation, which is the core of this work.** **Also, generalization errors do not deal with distributional shifts;** instead, they characterize the gap between training error and test error on the same distribution. In this work, we are aiming to measure the discrepancy between distributions to evaluate how well the given dataset is aligned with the target dataset. **The usage of these measures for deriving generalization bounds is principally different from what is needed in this work.**

---

> ### Author Response · Authors · 2022-11-14
> **Comment: "...Even though this data valuation metric is difficult to compute, the paper points out that the difference can be calculated efficiently, allowing for an efficient ranking of the data points..."**
>
> > *"The paper proposes a learning-agnostic data valuation framework based on optimal transport (OT). The authors suggest using a Class-wise Wasserstein distance between two datasets as a surrogate for validation performance, thus alleviating the need to pre-specifying the learning algorithm in advance. The authors then propose a new data valuation metric based on the gradient of the OT distance with respect to the perturbation on the probability mass associated with each data point. **Even though this data valuation metric is difficult to compute, the paper points out that the difference can be calculated efficiently, allowing for an efficient ranking of the data points.** The paper then presents empirical results to demonstrate the efficiency of the proposed methods on several different problems."*
>
> ***TL;DR:*** *The data valuation metric can be directly obtained for free from the output of the OT-solver without any additional computation. The difference can be calculated in a way that the error from entropy-regularization can be eliminated.*
>
> **Re:** The authors would like to thank the reviewer for providing helpful feedback and comments on the manuscript. We would like to first clarify a slight misperception.
>
> The original formulation is optimal transport (OT) is linear programming (LP), which may have a high computational complexity for practical problems. Mostly, people solve the entropy-penalized formulation (entropy-OT) instead, which can be solved quite efficiently with the iterative Sinkhorn algorithm. As we reported in Appendix B.5 "SCALABILITY EXPERIMENT" and Figure 17, the state-of-the-art solver for entropy-OT achieves a near-linear time complexity. Inevitably, **entropy-OT is an approximation to the original OT, and errors under certain thresholds are expected.** The error margin can be made arbitrarily small at the cost of increased computation complexities. **In practice, this is not considered a hard tradeoff as usually, one can achieve quite high accuracy (small errors from entropy regularization) before the computation becomes intractable.** For example, throughout this work, we set the entropy regularizer to 0.1. We observe no major difference in experiment results if these parameters are set lower or higher, thus we do not consider these as a significant factor.
>
> In this work, we propose to use the calibrated gradient as the metric for data valuation, which can be directly obtained from the output of off-the-shelf OT solvers without any additional computation. Since the calibrated gradient is based on the dual solutions to the entropy-penalized OT, it also inherits the errors from the entropy regularization. **Despite the fact that the errors from entropy regularization are not considered a prominent issue, we further show that in this specific scenario of data valuation, the errors can be completely dodged.**
>
> In data valuation, usually, it is the ranking of the values (which data are relatively "good" and which are relatively "bad") matter rather than the values themselves. **We present a method that calculates the clean ranking that is completely free from the errors from entropy regularization. It shed light on directions for new possibilities**–for example, we may be able to set a large error margin (strong entropy regularization) to speed up the solution for the entropy-OT while the quality of data valuation (the rankings) is totally unaffected.
>
> We are sorry for causing the confusion and the expressions in the manuscript have been modified accordingly.

---

> > ### Comment · Reviewer_5BsF · 2022-11-19
> > **Response**
> >
> > Since you are considering discrete measures $\mu_t$ and $\mu_v$, OT is a linear program and can be solved with off-the-shelf LP solvers.
> >
> > Am I understanding this correctly? Suppose $\mu_t$ and $\mu_v$ are not discrete, would the OT between them still be a linear program?

---

> > > ### Author Response · Authors · 2022-12-13
> > > **discrete measures, LP, solvers, and continuous OT**
> > >
> > > **a. Are you considering the distributions of the target and given datasets as discrete measures such that the OT problem is formulated as a linear program that can be solved with off-the-shelf LP solvers?\
> > > b. Does the OT problem remain a linear program if these distributions are not discrete?**
> > >
> > > > “...considering discrete measures \mu_t and \mu_v, OT is a linear program and can be solved with off-the-shelf LP solvers…. Suppose \mu_t and \mu_v are not discrete, would the OT between them still be a linear program? ”
> > >
> > > **Re: a.** Yes, the empirical distributions of the datasets are discrete measures and the corresponding OT problem is formulated as LP. To accelerate the solution procedure further and take advantage of parallel computing, we use off-the-shelf OT solvers that solve the entropy-regularized OT on GPU via the iterative Sinkhorn algorithm.
> > >
> > > **b.** Usually, only empirical distributions obtained from the data are available, hence only discrete measures are considered. Continuous OT is a much less explored topic. The solution procedure is more complicated and remains an active research area.
> > >
> > > Rigorous derivations and detailed discussions can be found in the following reference.
> > >
> > > > Peyré, G., & Cuturi, M. (2019). Computational optimal transport: With applications to data science. Foundations and Trends® in Machine Learning, 11(5-6), 355-607.
> > > https://arxiv.org/pdf/1803.00567.pdf
> > >
> > > This book provides a comprehensive tutorial for the modern usage of optimal transport in machine learning and fully covers the issues of interest in this discussion.
> > >
> > > Additionally, these slides also contain a nice introduction to these topics and may be helpful.
> > >
> > > > https://remi.flamary.com/biblio/presvannes2016.pdf \
> > > P9: general formulation with continuous distributions;\
> > > P10-11: empirical distribution with discrete measures;\
> > > P12 and after: entropic regularization

---

> ### Author Response · Authors · 2022-11-16
> **A friendly reminder**
>
> Dear Reviewer 5BsF,
>
> We would like to thank You for Your comments and suggestions, we tried our best to address every question raised. We hope that our answers could resolve Your concerns. We are happy to address additional suggestions. Since the rebuttal period is closing soon, we would love to be able to respond any further questions.
>
> Cordially,
>
> Authors of Paper857

---

> ### Author Response · Authors · 2022-11-19
> **Rebuttal period ending–we anticipate your feedback!**
>
> Dear Reviewer 5BsF,
>
> As the rebuttal/author discussion period is closing, we sincerely look forward to your feedback. The authors are deeply appreciative of your valuable time and efforts spent reviewing this paper and helping us improve it.
>
> It would be very much appreciated if you could once again help review our responses and let us know if these address or partially address your concerns and if our explanations are heading in the right direction.
>
> Please also let us know if there are any further questions or comments about this paper. We strive to consistently improve the paper and it would be our pleasure to have your precious feedback!
>
> Kind Regards,\
> Authors of Paper857

---

> > ### Comment · Reviewer_5BsF · 2022-11-19
> > **Thank you**
> >
> > Dear Authors of Paper857,
> >
> > Thank you for the detailed response, it helps a lot in clarifying some of the issues I had previously. Please consider incorporating some of your responses here (particularly the part about "radius for accurate prediction") into the manuscript as you see fit.
> >
> > I'd be happy to increase my score to 8. I also apologize for my late response. I would like to emphasize that my follow-up questions are for you to improve the paper, you do not need to respond to them.
> >
> > Take care,
> > Reviewer.

---

### Official Review · Reviewer_BSJ2 · 2022-10-23

**Confidence:** 4
**Clarity, Quality, Novelty And Reproducibility:** 1. In the introduction, one motivatio…
**Correctness:** 3
**Technical Novelty And Significance:** 3
**Empirical Novelty And Significance:** 4
**Recommendation:** 8

**Strength And Weaknesses:**

Advantages:
1. The paper is well-written and easy to follow. The authors additionally give lots of interpretations after results or formulations to help with the understanding.
2. Label information is incorporated into the dataset distance design.
3. The theoretical part of the paper is sound. Also, the connection from OT distance sensitivity to individual datapoint value is neat and natural.
4. The author provides rather extensive empirical validations of the method and showed state-of-the-art performance. An interesting new application on “irrelevant data detection” is also included, which was originally often discussed as mislabeled data detection.
5. Discussions on the limitations of LAVA in the conclusion are valid and indeed insightful.

Disadvantages:
1. The motivation for developing a learning-agnostic valuation method, especially in the context of the existing works with similar dataset distance approaches could be improved.
2. Some small parts of the experiment details require clarification.
For more details, see below.

**Summary Of The Paper:**

This paper proposes a datapoint valuation method, LAVA, that does not require a pre-defined learning algorithm, which is a common assumption in the existing literature. It utilizes the Wasserstein distance between the training set and the validation set with respect to a hybrid cost that considers both feature and label distances. The authors prove that the distance has theoretical connections with the validation performance (i.e., the utility) and show that the distance can be efficiently calculated using existing solvers. With that, the authors propose to use calibrated gradient to account for a datapoint’s contribution to the distance and thus its value. The paper has done extensive empirical experiments on 5 important application scenarios and demonstrated superior performances. Ablation studies are also done to better understand the behaviors of the proposed method.

**Summary Of The Review:**

Overall, I like the idea of a label-aware dataset distance measure as a data valuation metric despite borrowed from the OT literature. The method is backed by theoretical connections to validation performance, which is of interest in the topic of data valuation. The datapoint’s sensitivity to OT distance also naturally fits the marginal contribution concepts often used in data valuation. If my concerns above can be addressed, I think this work could contribute value to the community.

---

> ### Author Response · Authors · 2022-11-14
> **(minor) How did you implement the model trained on validation dataset to detect bad data?**
>
> > *"For Appendix B.2.4, I am interested to know how does the model directly trained on the validation dataset detect bad data?"*
>
> **Re:** For the model trained on validation dataset to detect bad data, we use that model to naively get the training data points’ confidence scores of their given class label and rank according to those scores (high confidence score means the model is more confident on that point for the given label which shows a higher chance of being good data).

---

> ### Author Response · Authors · 2022-11-14
> **(minor) How is Figure 1 created?**
>
> > *"Referring to the discussion on Figure 1, it would be helpful to briefly explain how do you select (or manipulate) the datasets to create different Wasserstein distances to the validation set when drawing the curves? Or alternatively, do you change the validation set?"*
>
> **Re:** We are sorry for the lack of clarification. In Figure 1, to achieve diverse levels of model performances, we have randomly flipped the labels in the full training dataset, keeping the validation dataset intact. Then we calculated the class-wise Wasserstein distance for each of them. As shown in Figure 1, even for different datasets and different models, we can observe a negative correlation between the model performance and the Wasserstein distance. We are sorry for this omission and will include these details in the updated version.

---

> ### Author Response · Authors · 2022-11-14
> **Why LAVA does not perform in Poisoning Attack Detection as well as in Backdoor Detection?**
>
> > *"Regarding the Poisoning Attack Detection experiments, you mentioned that this task is “especially hard to detect”. However, the baseline methods actually work better than those in Backdoor Detection. In contrast, LAVA does not perform here as well as that in Backdoor Detection. Any explanation?"*
>
> **Re:** There are two main reasons for LAVA to perform poisoning attack detection not as well as during the backdoor attack detection. Firstly, unlike the backdoor attack experiment, **in the poisoning case, the labels of the corrupted samples are not changed, which makes it harder to detect**, since LAVA cannot exploit label information in the distance computation. Secondly, **this attack has poisoned very few samples (only 50 samples) unlike the backdoor attack (2500 samples)**, which makes it more difficult for LAVA to find those samples through distributional distance.

---

> ### Author Response · Authors · 2022-11-14
> **Why some baselines perform worse than random guessing. Are they implemented correctly? Attack model accuracies do not start from the origin.**
>
> > *"In Figure 3, it really confuses me that most of the baseline methods perform even worse than random guessing in terms of the detection rate. Are the baselines implemented correctly? Or do you have an explanation for this abnormal behavior? Also, it is weird that some attack/model accuracies do not start from the same point when you start to throw away data (bottom plots)."*
>
> **Re:** We appreciate the observation and we have doubled checked that the baselines are indeed correctly implemented. We add an additional discussion for the performance of baselines here.
>
> **Influence functions (INF)**: this technique is known for being *fragile* [1,2,3] when applied to deep neural networks due to several reasons:
> + Influence function approximation formula assumes the linearity and convexity while neural networks are typically nonlinear and non-convex;
> + Moreover, the influence function assumes the model is trained to converge while early stopping is frequently used for training deep neural networks.
>
> **TracIN**: For the backdoor experiment, the two TracIn variants exhibit similar trends as influence functions, which only start finding the adversarially perturbed data points until filtering out a certain amount of clean data. We conjecture that this is because adversarially perturbed data points are designed to change the model prediction of particular target examples; each individual perturbed data point, however, does not have a significant impact on the loss of clean validation data. **Besides, adversarially perturbed data points are similar and their influences on validation data are also similar in nature, so they tend to be ranked together by influence-based methods.** For other settings, TracIn also does not perform well as the computation of these influence scores has a small signal-to-noise ratio due to random mini-batch selection [2].
>
> **KNN-SV**: This approach would perform well when the "bad" data point has a large euclidean distance from *both* the "good" data as well as other "bad" data. For backdoor and poison data detection, the "bad" data has a small distance from the "good" data. For noisy feature experiment, the distances within the "bad" data points are actually closer to each other, which will form a small cluster, as a result harming the performance of KNN-SV. For mislabeled data experiment, our interpretation for the poor performance of KNN-SV on CIFAR-10 is that the images from the same class may not necessarily be close due to the high-dimensionality nature.
>
> We apologize for the confusion. The attack/model accuracies in Figure 3 did not start from 0, which might be misleading. We updated the figure by drawing plots starting from 0 data removal.
>
> > [1] Basu, Samyadeep, Phil Pope, and Soheil Feizi. "Influence Functions in Deep Learning Are Fragile." International Conference on Learning Representations. 2020. \
> > [2] K, Karthikeyan and Anders Søgaard. “Revisiting Methods for Finding Influential Examples.” ArXiv abs/2111.04683 (2021). \
> > [3] Bae, Juhan, et al. "If Influence Functions are the Answer, Then What is the Question?." Neural Information Processing Systems, 2022.

---

> ### Author Response · Authors · 2022-11-14
> **Why is the feature extractor trained with the validation dataset? Will this introduce any form of bias since the validation dataset will be used for the data valuation step? Will an extractor trained on the overall training dataset work?**
>
> **Re:** Here, we can interpret the validation dataset as follows. There is a buyer who wants to purchase some data. They have their representative dataset, for which they want the purchased data to align with/perform well on. Then we can think of the purchased data as the training dataset and the representative dataset as the validation data. Given this scenario, the feature extractor is trained on the validation dataset for two reasons: i) the buyer would want to see the purchased data in terms of their (w.r.t.) their own representative data, ii) training the extractor on other data, which might also be bad data, would create bias toward undesired direction. For minimal and necessary requirements, it is fair to train the feature extractor on validation dataset from the buyer’s perspective. At the same time, the buyer only possesses their own data and it is not clear why training the feature extractor on the overall training dataset would be advisable. We agree with the reviewer that the feature extractor can be biased by the validation data and given the adversary who knows such bias might create examples to exploit the extractor. We think that a thorough study of data valuation in adversarial cases is an interesting future work.

---

> ### Author Response · Authors · 2022-11-14
> **Can you also clarify why is the difference in groundtruth values of the calibrated gradients in Section 3.2 enough to rank all data points? Do you calculate this difference with respect to a common datapoint selected and then rank them accordingly?**
>
> **Re:** Yes. The ranking is relative, so the selection of the baseline datapoint is irrelevant.

---

> ### Author Response · Authors · 2022-11-14
> **How to calculate the actual change in the OT distance when perturbing the probability mass? Given finite samples.**
>
> > *'In Section 3.1, it is unclear to me how to calculate the actual change in the OT distance when we perturb the datapoint probability mass by a given amount, if we calculate OT distance based on finite samples?'*
>
> **Re:** When given finite samples, we are dealing with (empirical) discrete distributions rather than continuous distributions, which is so much easier. **We can calculate the new discrete distribution after the change and re-calculate the OT distance from scratch.**

---

> ### Author Response · Authors · 2022-11-14
> **Can LAVA do subset selection, other than just point removal?**
>
> > *"The paper has extensive applications and great performance in detecting “bad” data. However, we care about “good” data as well. Can LAVA perform data summarization tasks (another common application of data valuation) as well? I know that Appendix B.4 is relevant to this question, but I am looking at a much smaller subset size for summarization. For example, can you select 1K points to train a network? This is essentially about the effectiveness of LAVA in picking representative and highly valued datapoints. I would expect a faster increase in model performance when including the highest value points first. "*
>
> **Re:** We appreciate the suggestion for the important application and have tested a data summarization experiment on CIFAR-10, where we select a subset of 500 highest value points and increment it by another 500 next highest values, subsequently. **As shown in Figure 16 of the updated version, our method performs better than the random selection and can find training points more effectively than other baselines, which in turn results in a better validation performance improvement.** However, even for this case, we would recommend continuous point removal from the dataset for a more exact summarization. We have updated the paper with a paragraph on this experiment, accordingly.

---

> ### Author Response · Authors · 2022-11-14
> **(References) MMD has been used already for data valuation or generalization gaps. How LAVA offers insights on the basis of existing works?**
>
> > [1] Tay, Sebastian Shenghong, et al. "Incentivizing collaboration in machine learning via synthetic data rewards." Proceedings of the AAAI Conference on Artificial Intelligence. Vol. 36. No. 9. 2022.\
> > [2] Wu, Zhaoxuan, Yao Shu, and Bryan Kian Hsiang Low. "Davinz: Data valuation using deep neural networks at initialization." International Conference on Machine Learning. PMLR, 2022.\
> > [3] Villani, Cédric (2009). Optimal Transport, Old and New. Grundlehren der mathematischen Wissenschaften. Vol. 338. Springer-Verlag Berlin Heidelberg. p. 10. doi:10.1007/978-3-540-71050-9. ISBN 978-3-540-71049-3\
> > [4] Kullback, Solomon, and Richard A. Leibler. "On information and sufficiency." The annals of mathematical statistics 22.1 (1951): 79-86.\
> > [5] Rudin, Leonid I., Stanley Osher, and Emad Fatemi. "Nonlinear total variation based noise removal algorithms." Physica D: nonlinear phenomena. 1-4 (1992): 259-268.\
> > [6] Gabor J Szekely, Maria L Rizzo, et al. Hierarchical clustering via joint between-within distances: Extending ward’s minimum variance method. Journal of classification, 22(2):151–184, 2005\
> > [7] David Alvarez-Melis and Nicolo Fusi. Geometric dataset distances via optimal transport. Advances in Neural Information Processing Systems, 33:21428–21439, 2020.\
> > [8] Jean Feydy, Thibault Séjourné, François-Xavier Vialard, Shun-ichi Amari, Alain Trouvé, and Gabriel Peyré. Interpolating between optimal transport and mmd using sinkhorn divergences. In The 22nd International Conference on Artificial Intelligence and Statistics, pages 2681–2690. PMLR, 2019.\
> > [9] Aude Genevay, Gabriel Peyré, and Marco Cuturi. Learning generative models with sinkhorn divergences. In International Conference on Artificial Intelligence and Statistics, pages 1608–1617. PMLR, 2018.

---

> ### Author Response · Authors · 2022-11-14
> **MMD has been used already for data valuation or generalization gaps. How LAVA offers insights on the basis of existing works?**
>
> > *"This is related to the novelty of the work, and how insight it has contributed to the community. There have been attempts to use distribution divergence or dataset differences for data valuation. One reason for choosing OT is that OT is computationally tractable from finite samples: As far as I know, MMD used in [1] has similar properties. It is also essential to draw connections between distribution divergence to learning performance: Another work [2] also partially uses MMD to bound generalization performance. Discussions might be needed to examine the significance of LAVA in the context of the existing works mentioned above."
> > \
> > [1] Incentivizing Collaboration in Machine Learning via Synthetic Data Rewards. AAAI 2022.\
> > [2] DAVINZ: Data Valuation using Deep Neural Networks at Initialization. ICML 2022.*
>
> **Re:** We appreciate the reviewer mentioning other divergence metrics as we would like to underline the importance of our work in comparison with those measures. There are, indeed, other discrepancy measures, such as Kullback-Leibler Divergence [4], Total Variation norm [5], or Maximum Mean Discrepancies (MMD) [1,2,6], that may work similarly to the optimal transport/Wasserstein distance and be able to bound the validation errors. As has already been noted, the mathematically well-defined OT distance has advantageous analytical properties (is a valid metric; compatible with sparse-support distributions; stable with respect to deformations of the distributions’ supports), and it is also computationally tractable and commutable from finite samples [3,7,8]. In addition, we provide here more discussions and direct comparisons with MMD.
>
> As the reviewer kindly pointed out, the MMD norm has seen extensive usage in tasks such as domain adaptation, deriving learning bounds, or generative models. The popular MMD metric may be considered a proximity to OT/Wasserstein distance as for analytical properties. Notably, MMD is also computable from finite samples, has a smaller sample complexity, and is cheaper to compute when compared to OT. With all of the said similarities, though, we note that there are, still, considerable differences in their theoretical properties, which lead to major distinctions in the development of downstream methods.
>
> First, MMD does not faithfully lift the ground distance to the space of probability measures. For example, the discrepancy calculated by OT between a reference measure and its translation will be exactly the distance of this translation; **while for MMD, the measured discrepancy will be distorted due to the convolutions in the Fourier transform of the kernel function (if using a characteristic kernel, such as the commonly used squared exponential kernel as in [1]), where the extent of distortion strongly depends on the smoothness of the reference measure.** As per [7], in practice, this theoretical disadvantage for the MMD norm may result in vanishing gradients or other defects near the extreme points of the measures' support. The reference provides detailed discussions of the issue as well as other comparisons between OT and MMD.
>
> In the scenario of dataset/datapoint valuation, we are mostly dealing with empirical distributions from finite samples. The theoretical limitations may constrain the possibilities for downstream applications. Besides, as can be seen in [1], significant efforts are needed to find an appropriate kernel function, and compromises are necessary for conditions to be met. **We note optimal transport has better analytical properties as the discrepancy measure for this use case.**
>
> Besides, optimal transport is formulated as linear programming, which naturally lends itself to performing sensitivity analysis w.r.t. the parameters of interest, allowing us to extend it to the application of individual data valuation. Specifically, **the OT formulation allows us to obtain the gradients w.r.t. the probability mass for free from the already available dual solutions.** Also, due to the linear nature of the formulation, these gradients are precise within a non-trivial range and thus making it possible for us to reliably predict the contribution of each datapoint.
>
> **As far as we know, these properties do not directly apply to the MMD norm. As also witnessed in existing work [1,2], MMD-based valuation methods can evaluate datasets but cannot directly evaluate single data points; instead, they calculate the individual data valuation as its marginal contribution to the dataset containing the point. However, as implemented in [2] that would require (after speed-ups and batch-wise approximations) around 10 seconds for evaluating each point in CIFAR10. This deems the valuation of all datapoints completely unattainable to finish in a reasonable time.**
>
> **We believe this gives optimal transport an important edge in the comparison as it requires magnitudes fewer evaluations to obtain the valuation for each datapoint.**
>
> (References are in the response below.)

---

> ### Author Response · Authors · 2022-11-14
> **(2/2) Comment 1**
>
> (Cont'd)
>
> > *"...LAVA performs a LOO-like computation and does not consider combinations (like in CGT)...*
>
> **c.(i).** **Compared to LOO, LAVA is model-free and LAVA directly predicts the LOO-like results without the need to conduct repeated calculations for every single datapoint as in LOO.**
>
> **c.(ii).** All the results yielded from LAVA are relative to the entire dataset. No subset-wise internal combinatorial effects are considered. We note this to be where LAVA's remarkable computational advantages are from. **More importantly, the combinatorial effects considered in CGT do not necessarily provide advantages in practical applications–LAVA demonstrated strong and impressive empirical results.**
>
> > *"...Give formal justifications with references that the LOO and CGT methods “remain expensive” given the approximations."*
>
> **d.(i).** On the theoretical front, principally, **LOO requires retraining the model on every dataset with one datapoint removed.** For modern ML models and datasets, each model retraining may take a considerable amount of time. **Even for approximations that use proxy models to simplify the retraining process, LOO still needs to be computed repeatedly for each datapoint.** Similarly, CGT typically requires an exponential number of retraining for the evaluation of each datapoint. Even with state-of-the-art approximations that may rely on Monte Carlo sampling to reduce the number of retraining or use proxy models [3,4,5] to simplify the evaluation, it remains high in computational complexity in either inference or training time. and is generally more demanding than LOO.
>
> **LAVA only computes a single optimal transport problem, which scales almost linearly with the size of datasets leveraging state-of-the-art OT solvers (Figure 17, Appendix B.6). Its computational complexities are orders of magnitudes less than LOO and CGT.**
>
> **d.(ii).** We have included the verification of these claims in the empirical results. Other than the scalability discussion mentioned above, we also provided comparisons of actual computational times with prevailing LOO and CGT methods. As depicted in Figure 5 in Section 5 "Related Work", **it is immediately noticeable that LAVA is impressively faster than all comparable methods.** Despite the fact that the exact computation times may vary under different implementation settings, it is safe to conclude that LAVA is lightning-fast compared to existing approaches and in some way, "addressed" the notorious issue of computational intractability in data valuation. **LAVA is novel for the methods of its kind and marks its distinct contribution in the computational aspect.**
>
> References:
> > [1]  Basu, Samyadeep, Philip Pope, and Soheil Feizi. "Influence functions in deep learning are fragile." arXiv preprint arXiv:2006.14651 (2020). \
> > [2] K, Karthikeyan and Anders Søgaard. “Revisiting Methods for Finding Influential Examples.” ArXiv abs/2111.04683 (2021). \
> > [3]  Covert, Ian, and Su-In Lee. "Improving KernelSHAP: Practical Shapley value estimation using linear regression." International Conference on Artificial Intelligence and Statistics. PMLR, 2021. \
> > [4] Jia, Ruoxi, et al. "Efficient Task-Specific Data Valuation for Nearest Neighbor Algorithms." Proceedings of the VLDB Endowment 12.11. \
> > [5] Jethani, Neil, et al. "FastSHAP: Real-Time Shapley Value Estimation." International Conference on Learning Representations. 2021.

---

> ### Author Response · Authors · 2022-11-14
> **(1/2) Comment 1**
>
> **a. Does LAVA really address the computational issue of LOO and CGT? \
> b. LAVA is essentially like a perturbation-based contribution measure similar to influence function.\
> c. LAVA performs a LOO-like computation and does not consider combinations (like in CGT).\
> d. Give formal justifications with references that the LOO and CGT methods “remain expensive” given the approximations.**
>
> > *"In the introduction, one motivation for a learning-agnostic method is that the retrainings of the models are too expensive in LOO and CGT. However, I do not think LAVA directly addresses this problem, at least about CGT, because LAVA is more like a perturbation-based contribution measure essentially similar to influence function (INF). LAVA performs a LOO-like computation and does not consider combinations (like in CGT). Also, please give formal justifications with references that the LOO and CGT methods “remain expensive” given the approximations."*
>
> **Re:** Thanks for posing these detailed and intriguing questions. We would like to take this chance to elaborate on how the proposed method compares to existing approaches in both methodology and computational aspects and clarify potential ambiguities that we may have not sufficiently discussed.
>
> > *"In the introduction, one motivation for a learning-agnostic method is that the retrainings of the models are too expensive in LOO and CGT. However, I do not think LAVA directly addresses this problem, at least about CGT..."*
>
> **a.** **LAVA only computes a single optimal transport problem (solved as convex optimization with efficient algorithms) to obtain all data valuation results.** LOO requires retraining the model on every dataset with one datapoint removed while CGT typically requires exponentially more retraining than LOO. *(More detailed and technical comparisons are provided below.)*
>
> > *"...LAVA is more like a perturbation-based contribution measure essentially similar to influence function (INF)...*
>
> **b.(i).** **INF is inherently model-based where its linearity assumptions for perturbation analysis mismatch with the nonlinearity nature of models, rendering the method particularly fragile [1].** LAVA is based on optimal transport which is formulated as Linear Programming (LP), where the linear assumptions naturally hold and enable the method to be notably robust across scenarios. Thus, INF results are never accurate for nonlinear models (which are, practically, almost always the case) and may have large deviations depending on the model [2]; **LAVA results are accurate in a non-trivial range of perturbations and remain close approximations beyond the range.**
>
> **b.(ii).** INF needs to be calculated repeatedly for each datapoint while LAVA solves a single optimal transport problem to obtain valuation results for all datapoints. **LAVA is magnitudes more efficient and scalable.**

---

> ### Author Response · Authors · 2022-11-16
> **A friendly reminder**
>
> Dear Reviewer BSJ2,
>
> We want to thank You for Your helpful comments, which led to a number of interesting discussions and comparisons. We have responded to each of Your concerns and questions. Hopefully, You will find that they adequately address Your concerns. Before the rebuttal phase is over, please let us know if You have any more questions or need any clarification. We would be happy to address them.
>
> Best Wishes,
>
> Authors of Paper857

---

> > ### Comment · Reviewer_BSJ2 · 2022-11-18
> > **Post rebuttal**
> >
> > I would like to thank the authors for their detailed responses. The clarification of the OT method is essential to me. I appreciate the author's effort in providing justifications and additional experiments. My concerns are addressed and I have increased the score.

---

> > > ### Author Response · Authors · 2022-11-19
> > > **We appreciate your help in improving this paper**
> > >
> > > Dear Reviewer BSJ2,
> > >
> > > The authors extend their deep appreciation to the reviewer for sparing valuable time and effort reviewing this paper and helping us improve it. Your valuable feedback has a profound impact on the development of this paper and also on our understanding of this paper and broader issues. Your insights and comments have influenced the trajectory of our line of research as well as the development of our future work.
> > >
> > > In all, we would like to express our gratitude to the reviewer for being with us during this process and it is our pleasure to have this valuable experience!
> > >
> > > Kind regards,\
> > > Authors of Paper857

---

### Official Review · Reviewer_uRtv · 2022-10-25

**Confidence:** 3
**Correctness:** 4
**Technical Novelty And Significance:** 3
**Empirical Novelty And Significance:** 3
**Recommendation:** 8

**Clarity, Quality, Novelty And Reproducibility:**

While the paper applies the existing Wasserstein distance between datasets to the data valuation problem, it proposes novel perspectives through the new insight into the connection between the distance and the validation performance and the use of gradients in constructing an efficient valuation method. The paper is well-written and easy to follow. The significance of the proposed approach is demonstrated through many use cases in the experiments.


**Strength And Weaknesses:**

The strength of the paper is in the application of the existing Wasserstein distance between datasets to the data valuation problem with the new insights on the connection between the distance and the validation performance. Additionally, it demonstrates many use cases of the proposed data valuation method in the experiments. The paper is also well-written so that readers can understand the main idea easily.

The main weakness of the paper is probably due to the fact that the proposed data valuation only works for individual data points. I have several questions as follows.

1. The proposed approach requires a validation dataset with labels. How would the Wasserstein distance be modified to measure the distance between a training set and a 'reference dataset' without labels?

2. While the paper motivates the problem of data valuation as a central method to the emerging data economy (in data exchange), the proposed method cannot value data sources (which consist of multiple data points). Hence, I am wondering how this approach can be applied in data exchange (between multiple organizations/companies with different datasets).

3. It is often the case that there are duplicates of data points (or redundant data points that are very similar to one another) in the real-world dataset. For simplicity, let us consider a dataset consisting of 3 data points x1,x2,y where x1 and x2 are duplicates of each other; and y is different from x1,x2. Intuitively, removing x1 does not affect the value of the dataset (since x2 contains the same information as x1), so the value of x1 is low. Hence, point-wise values of x1 and x2 are low. However, if we remove both x1 and x2 (since they have low valuation), then it is problematic because y does not contain information in x1 and x2. How does the proposed pointwise data valuation method work in this case?

4. Like the work of Koh & Liang (2017), can the proposed method be used to explain a particular prediction of the model by finding the training data point that is responsible for the prediction? In this case, the analogous notion of a validation set contains only a single data point (the prediction to be explained), so I am wondering if the approach can still work.


**Summary Of The Paper:**

The paper introduces a data valuation method of individual training data points that does not requires learning algorithms via the gradient of the Wasserstein distance between the training set and validation set w.r.t. the perturbations on the probability mass of a data point. While it is based on existing work on the hierarchically-defined Wasserstein distance between datasets, the paper provides new insight into the connection between this distance and the validation performance which is crucial for data validation. Furthermore, it proposes an efficient data valuation method of individual training data points based on the sensitivity of this distance, which has a wide range of use cases as demonstrated in the experiments.

**Summary Of The Review:**

While limited to pointwise data valuation, this work proposes a new perspective in constructing a data valuation method without a learning algorithm and empirically evaluates the performance in various use cases. Thus, I believe the contribution and quality of the paper are sufficient.

---

> ### Author Response · Authors · 2022-11-14
> **Can LAVA explain a particular prediction of the model by finding the training data point that is responsible for the prediction?**
>
> > *"Like the work of Koh & Liang (2017), can the proposed method be used to  In this case, the analogous notion of a validation set contains only a single data point (the prediction to be explained), so I am wondering if the approach can still work."*
> >
> **Re:** We would like to note that our method is not directly created for this task. One could try using the transport map from the LAVA solution to detect training points with the closest connection to the data point of interest, which can be thought of as the most responsible points for the prediction. **However, there are direct lines of work that focus on this specific setting, such as the mentioned Influence Function [1] or Memorization and Influence Estimation [2].**
>
> > [1]. Koh, P. W., & Liang, P. (2017, July). Understanding black-box predictions via influence functions. In International conference on machine learning (pp. 1885-1894). PMLR.  \
> > [2]. Feldman, V., & Zhang, C. (2020). What neural networks memorize and why: Discovering the long tail via influence estimation. Advances in Neural Information Processing Systems, 33, 2881-2891.

---

> ### Author Response · Authors · 2022-11-14
> **For mutually redundant (e.g. duplicated) data points, how to define their value in an equitable and practically meaningful manner?**
>
> > *"It is often the case that there are duplicates of data points (or redundant data points that are very similar to one another) in the real-world dataset. For simplicity, let us consider a dataset consisting of 3 data points x1,x2,y where x1 and x2 are duplicates of each other; and y is different from x1,x2. Intuitively, removing x1 does not affect the value of the dataset (since x2 contains the same information as x1), so the value of x1 is low. Hence, point-wise values of x1 and x2 are low. However, if we remove both x1 and x2 (since they have low valuation), then it is problematic because y does not contain information in x1 and x2. How does the proposed pointwise data valuation method work in this case?"*
>
> **Re:** The reviewer has brought up an interesting case that is related to the **fundamental difference between data valuation and data selection**, which will be detailed later.
>
> First, we agree that removing a duplicate does not change the value of dataset (which is confirmed through Table 3 in Appendix B.9, Page 21) and that the two duplicates will receive the same value using our method.
>
> It is argued in prior literature that **a fundamental property that needs to be respected by data valuation is fairness**, i.e., giving the same value to two data points that contribute the same. Hence, in fact, not only LAVA but all recent fair data valuation methods will assign the same value to two identical data points. However, if one uses the values calculated by these methods to remove data, two identical points will inevitably be removed at the same time, which is clearly not an optimal data selection strategy that can maximize learning performance. Hence, there is a fundamental difference between fair data valuation and data selection, in the sense that fair data valuation is not the optimal solution to a data selection problem in general cases, especially when there exist duplicates in the dataset. On the other hand, data valuation provides a nearly optimal solution when the dataset contains some bad data, such as mislabeled data and poisoned samples, but does not contain duplicates. The bad samples should all receive low values and thus removing them simultaneously is desirable which gives rise to a clean set that achieves the highest model performance.
>
> Despite the fundamental difference, data removal performance has often been used by the past literature to evaluate the quality of a data valuation method. Hence, this paper follows the same evaluation protocol. We argue that this evaluation protocol is still reasonable because the datasets used in both the past literature and our work are all well-curated and do not contain many duplicates; at the same time, the datasets are manually injected with some bad data (e.g., mislabeled), which makes data valuation a nearly optimal solution to removal. If the datasets do contain duplicates, then we will recommend two strategies to use the datasets to evaluate a data valuation technique through data removal: one way is to discard the duplicates before removal; the second way is to recalculate the values after each data point gets removed and then remove the next point based on new values. Both strategies will make LAVA as well as existing fair data valuation methods work in the case described by the reviewer.

---

> ### Author Response · Authors · 2022-11-14
> **How to evaluate multiple data sources?**
>
> > *"While the paper motivates the problem of data valuation as a central method to the emerging data economy (in data exchange), the proposed method cannot value data sources (which consist of multiple data points). Hence, I am wondering how this approach can be applied in data exchange (between multiple organizations/companies with different datasets)."*
>
> **Re:** From a computational efficiency perspective, *data point valuation* is arguably more challenging than *dataset valuation* because the number of data points is often orders of magnitude larger than the number of datasets combined in typical learning tasks. The original focus of LAVA is data point valuation, and we aim to address the severe computational challenge there.
>
> However, there are many ways to apply the proposed methodology to dataset valuation. One is to value each point in the combined datasets using LAVA and then calculate the value of each dataset by simply summing up the values of all points in the dataset. Another way is to directly calculate the proposed distributional distance between a target dataset and the validation set. However, the detailed comparison between these potential ways for dataset valuation in the context of data exchange might be worth a new paper, and we will defer it to future work.

---

> ### Author Response · Authors · 2022-11-14
> **Validation data without labels?**
>
> > *"The proposed approach requires a validation dataset with labels. How would the Wasserstein distance be modified to measure the distance between a training set and a 'reference dataset' without labels?"*
>
> **Re:** **Our setting follows the standard setting of prior literature on data valuation**, which aims at valuing labeled data. In fact, **labels are required by existing methods**, since they all value data based on the contribution to model performance, and label information is needed to evaluate model performance. **By contrast, our work can be easily extended to value unlabeled data.** In particular, our proposed utility function is a class-wise Wasserstein distance between two distributions; when the labeled information is not available, the class-wise Wasserstein distance can be simply reduced to the original Wasserstein distance between two feature distributions, disregarding all labels.

---

> ### Author Response · Authors · 2022-11-16
> **A friendly reminder**
>
> Dear Reviewer uRtv,
>
> We want to thank You for Your helpful comments, which led to a number of important extensions of our work. We have addressed each of Your questions. Please let us know if You have any more questions, we would be happy to address them within our allowed period.
>
> Kind Regards,
>
> Authors of Paper857

---

> ### Author Response · Authors · 2022-11-19
> **Rebuttal period ending–we anticipate your feedback!**
>
> Dear Reviewer uRtv,
>
> As the rebuttal/author discussion period is closing, we sincerely look forward to your feedback. The authors are deeply appreciative of your valuable time and efforts spent reviewing this paper and helping us improve it.
>
> It would be very much appreciated if you could once again help review our responses and let us know if these address or partially address your concerns and if our explanations are heading in the right direction.
>
> Please also let us know if there are any further questions or comments about this paper. We strive to consistently improve the paper and it would be our pleasure to have your precious feedback!
>
> Kind Regards,\
> Authors of Paper857

---

> ### Comment · Reviewer_uRtv · 2022-11-19
> **Thank authors for the detailed responses**
>
> I would like to thank the authors for the detailed responses which help to resolve my questions on the paper. In particular, I very much appreciate the discussion on data validation without labels and the difference between data valuation and data selection.

---

### Official Review · Reviewer_7VZM · 2022-10-25

**Confidence:** 3
**Correctness:** 4
**Technical Novelty And Significance:** 3
**Empirical Novelty And Significance:** 4
**Recommendation:** 8

**Clarity, Quality, Novelty And Reproducibility:**

The paper is very organized and well-written. The contribution is novel as far as I know.

**Strength And Weaknesses:**

 The paper studies an interesting problem and proposes a novel data valuation method using a natural measure based on the class-wise Wasserstein distance. They provide solid theoretical justification for their approach by proving that their measure characterizes the upper bound of the validation performance of any given models. They also propose an interpretable and easily-computable measure for individual data valuation. Finally, they conducted extensive experiments to test the efficacy and efficiency of their approaches in practical use cases.


**Summary Of The Paper:**

The paper proposes a data valuation method that is oblivious to the downstream learning algorithm. The main idea is to evaluate the training data by a class-wise Wasserstein distance between the training and the validation set. They prove that the class-wise Wasserstein distance approximates the performance for any given model under certain Lipschitz conditions. They also propose a method to evaluate individual data by the sensitivity analysis of this class-wise Wasserstein distance. Finally, the paper empirically evaluates the performance and efficiency of their methods. They show that their method improves the state-of-the-art performance while being orders of magnitude faster.


**Summary Of The Review:**

Overall, I think this is a good paper with nice conceptual contribution and solid theoretical analysis and extensive empirical evaluations that show the efficacy of their data valuation approaches in practical use cases.

---

> ### Author Response · Authors · 2022-11-14
> **Words of Appreciation**
>
> We would like to thank You for the positive assessment!

---

### Author Response · Authors · 2022-11-14
**Summary**

Dear Reviewers,\
We sincerely appreciate all Your insightful comments and questions. We are pleased that our work was recognized as a novel data valuation method with a solid theoretical justification (7VZM) that neatly and naturally connects to datapoint value (BSJ2) in efficient manner for significant applications (uRtv), and contributes to the accessible and applicable research for the community (5BsF,BSJ2).
To further improve understanding of our work, we have addressed the following aspects:

---
+ Comparison of our class-wise OT metric with other distributional measures, such as Maximum Mean Discrepancy (MMD), Kullback-Leibler Divergence (KL), or Total Variation norm (showing similarities and advantages in data valuation setting).
+ Highlighting the difficulty and importance of our theoretical contribution that fills the blank and validates our method.
+ Pointing out differences between LAVA against LOO and CGT-based methodologies.
+ Providing more insights into the reasoning behind the class-wise OT and its sensitivity analysis as a valid metric for dataset valuation as well as data point valuation.
+ Explanations for extension of LAVA to possible applications of our work: no label valuation, multiple contributors valuation, mutual redundancy, prediction explanation, and data summarization (including an experimental result).
+ Minor: figure explanations, experimental setup, clear figures, baseline performance explanations.

---
Please notify us if You have any further suggestions, we would be glad to discuss them.

With Gratitude,\
 Authors of Paper857

---

### Author Response · Authors · 2022-12-13
**A letter to the AC and all reviewers**

We sincerely appreciate the hard work of dedicated reviewers for helping review our manuscript  and providing helpful comments and valuable feedback. We are grateful for having you working alongside us during this journey.

**To catalyze future advances in the field, we will open-source our evaluation framework and provide a well-capsulated Python package.** The temporary code repository can be found at https://anonymous.4open.science/r/LAVA-data-value-2CE2 .We are developing a website to promote the package, providing tutorials and instructions as well as maintenance, updates, and technical support in the longer term. We will launch the website at the time this work is made public.

This work advances the field of data valuation **by significantly improving computational efficiency and enabling new application scenarios with only limited information about downstream learning algorithms.** Compared to the fastest baseline, our method reduces the time of data valuation on a typical task from more than 2 hours to less than 5 minutes. For most other methods that perform data valuation, the same experiment would take days or more–if it is possible to finish. It marks an essential step toward real-world implementations and has significant implications for both research and industrial applications.

**This work will have significant impacts beyond fundamentally advancing the research on data valuation.** The techniques developed in this work can be applied to a variety of other subfields of ML related to robustness, interpretability, data acquisition, etc. The results of this paper will facilitate the automation of data quality management in machine learning, which in turn, accelerates research and improves services based on ML. Data valuation is also at the heart of the global data economy. The advancements in data valuation made in this work will substantially benefit the development of data markets and promote data sharing, contributing to the business and economy as well as society as a whole.

We look forward to the discovery of further results down this line of research.

With best regards,\
Authors of Paper857

---

### Decision · Program_Chairs · 2023-01-20

**Decision:**

Accept: notable-top-25%

**Justification For Why Not Higher Score:**

This paper is a bit niche

**Justification For Why Not Lower Score:**

It's the paper with the hightest grade and, though a bit niche, could interest a wide audience.

**Metareview: Summary, Strengths And Weaknesses:**

This paper considers a new framework of data-valuation with interesting theoretical results.

All reviewers are very positive about this paper (with a perfect consensus at 8 which is quite uncommon). I share their opinion and can only recommend acceptance ! The contributions are novel and interesting.

**Note From Pc:**

if the above contains the word "oral" or "spotlight" please see: "oral" presentation means -> notable-top-5% and "spotlight" means -> notable-top-25%. As stated in our emails, we are disassociating presentation type from AC recommendations